# Effectiveness of clinical decision support in fall prevention among older adults: A systematic review and meta-analysis

Rune Solli[1]*, Nina Rydland Olsen[2], Linda Aimée Hartford Kvæl[1,3], Stijn Van de Velde[4], Are Hugo Pripp[1], Signe Agnes Flottorp[5], Therese Brovold[1]

**1** Department of Rehabilitation Science and Health Technology, Faculty of Health Sciences, OsloMet – Oslo Metropolitan University, Oslo, Norway, **2** Department of Health and Functioning, Western Norway University of Applied Sciences, Faculty of health and social sciences, Bergen, Norway, **3** Norwegian Social Research (NOVA), OsloMet – Oslo Metropolitan University, Oslo, Norway, **4** MAGIC Evidence Ecosystem Foundation; Research Department, Lovisenberg Diaconal Hospital, Oslo, Norway, **5** Division of Health Services, Norwegian Institute of Public Health, Oslo, Norway

* runesoll@oslomet.no

## Abstract

### Background

Systematic use of Clinical Decision Support (CDS), which provides timely information to assist healthcare practitioners in decision-making, is recommended in the implementation of fall prevention among older adults. This systematic review aimed to evaluate the effects of CDS for fall prevention on healthcare practitioners' adherence to recommended practice, medication outcomes, and patient outcomes.

### Methods

We searched Medline, EMBASE, CINAHL, Cochrane Library, Web of Science, AMED, PEDro, and Google Scholar from the earliest available dates through January 2025. We included randomised and non-randomised studies that directly compared interventions consisting of CDS presented on-screen or on paper to healthcare practitioners aiming to prevent falls in persons aged 65 years or older. We analysed healthcare practitioner performance, medication review and prescribing, fall risk, fall rate, and fall injury rate as primary outcomes. Two reviewers independently screened studies and assessed for risk of bias. We synthesised results using meta-analyses and vote-counting based on direction of effect, when possible, otherwise narratively, and we rated the certainty of the evidence using the GRADE approach.

### Results

Of 25 included studies, 20 were randomised and five were non-randomised. Most CDS tools supported healthcare practitioners in performing multifactorial fall risk assessments and follow-up interventions based on identified risks (60%) and most were

**Data availability statement:** All relevant data are within the manuscript and its Supporting information files.

**Funding:** This study was funded by the Research Council of Norway under grant number 301996, awarded to OsloMet – Oslo Metropolitan University, Faculty of Health Sciences. The funder did not play any role in the study design, data collection and and analysis, decision to publish, or preparation of the manuscript.

**Competing interests:** The authors have declared that no competing interests exist.

delivered electronically (60%). CDS may improve healthcare practitioners' adherence to recommended practice (all eight comparisons favouring CDS; 95% confidence interval [CI] 68% to 100%; low certainty) and likely improve medication review and prescribing (all nine comparisons favouring CDS; 95% CI 70% to 100%; moderate certainty), although the effect sizes are uncertain. CDS may reduce fall risk, but the effect may be small (odds ratio 0.93; 95% CI 0.81 to 1.01; low certainty). CDS likely reduces fall rates in hospitals or residential care (rate ratio [RaR] 0.74; 95% CI 0.63 to 0.88; moderate certainty) and in patients aged 80 years or older (RaR 0.72; 95% CI 0.61 to 0.86; moderate certainty). CDS may reduce fall rates in community-dwelling older adults (RaR 0.97; 95% CI 0.93 to 1.00; moderate certainty) and in patients aged between 65 and 80 years (RaR 0.92; 95% CI 0.84 to 1.01; low certainty), though the effects in both of these subgroups may be small. CDS may reduce fall injury rates in older adults aged between 65 and 80 years (RaR 0.80; 95% CI 0.59 to 1.09; low certainty). The evidence on fall injury rates in patients aged 80 years or older was very uncertain.

## Conclusion

CDS likely enhances healthcare practitioners' performance in fall prevention among older adults; however, the effect sizes remain unknown. Although CDS may improve patient outcomes in fall prevention, both the effect sizes and the certainty of evidence vary. Results from this study may inform the planning and implementation of CDS in fall prevention. Future studies should strive for clearer reporting of CDS design factors to allow for an evaluation of which factors may influence the success of CDS interventions in fall prevention.

## Trial registration

**Registration:** PROSPERO, CRD42021250500.

---

## Introduction

Falls are a major cause of morbidity and mortality among community-dwelling older adults aged 65 years or older [1–3]. Annually, one in three older adults experiences a fall, which can lead to significant health loss and increased care needs [1–5]. Globally, fall-related injuries are one of the most expensive conditions in terms of economic expenditures [4–6]. Identified risk factors for falls include a history of falls, higher age, female sex, and fear of falling [7,8]. A range of cost-effective interventions has been shown to prevent falls and reduce the incidence of fall-related injuries in older adults [9–11]. These interventions include muscle strength and balance training, home safety assessments and modifications, and medication adjustments [9,10]. Falls are often underreported, as older adults may not report falls unprompted [12,13]. The World Falls Guidelines 2022 (WFG2022) [14] emphasise the need to identify older adults at increased fall risk. They recommend that all older adults receive advise on fall prevention and physical activity.

Despite scientific support for the implementation of fall-prevention recommendations for older adults, the systematic uptake of evidence-based fall prevention practices has been slow [14–17] Consequently, fall rates and fall-related mortality have not declined [3,18,19]. The WFG2022 recommend the systematic use of Clinical Decision Support (CDS) in fall prevention to identify older adults at increased risk of falling and to facilitate fall risk assessments and interventions. CDS is an implementation strategy found to improve healthcare practitioners' adherence to clinical guidelines [20]. CDS is defined as computerised or non-computerised tools that combine health-related and medical information with individual patient information to support clinical decision-making [21,22]. CDS has the potential to assist healthcare practitioners in identifying older adults at increased fall risk [23–25] as well as in individualising fall prevention interventions based on identified risk factors [24,26–28].

Several systematic reviews have evaluated the effects of CDS used by healthcare practitioners across a variety of settings, demonstrating small to moderate positive effects [20,29–37]. Generally, the use of CDS may improve the implementation of clinical practice guidelines [20] and the performance of nurses and allied health professionals [29,34,35]. In addition, CDS has been shown to improve hospital care for older patients [30], medication outcomes in older adults [31,32], and medication outcomes in adults more broadly [36,37]. The effect of CDS on patient outcomes, however, appears less certain [35]. While CDS may not significantly affect mortality, it may moderately improve morbidity outcomes [33] and a variety of other patient outcomes [36]. Two systematic reviews found that CDS may help prevent falls in nursing homes [29] and in hospitals [30]. However, these reviews did not include several known studies on the effects of CDS in fall prevention [38,39], and no statistical syntheses were conducted. Overall, findings from systematic reviews indicate that CDS may provide important benefits for healthcare practitioners' performance and patient outcomes. However, no systematic review to date has specifically investigated the effects of CDS on healthcare practitioners' performance and patient outcomes in the context of fall prevention.

### Objectives

This systematic review aimed to evaluate the effects of CDS for fall prevention on healthcare practitioners' adherence to recommended practice, medication outcomes, and on patient outcomes.

## Materials and methods

### Protocol and registration

The review protocol was drafted and finalised in accordance with the Preferred Reporting Items for Systematic Reviews and Meta-Analyses (PRISMA) extension for study protocols (PRISMA-P) [40] and was registered in the PROSPERO international prospective register of systematic reviews (identification number CRD42021250500) prior to commencing the systematic review. We followed the recommendations in the Cochrane Handbook [41] and the guidance from the Cochrane Effective Practice and Organisation of Care (EPOC) [42]. Reporting of this systematic review adhered to the standards of the Preferred Reporting Items for Systematic Reviews and Meta-Analyses (PRISMA) [43]. See S1 Table for the populated PRISMA checklist. See S2 Table for differences between the protocol and the review.

### Eligibility criteria

We included studies evaluating the effects of CDS interventions designed to support HCPs in making decisions to prevent falls in older adults. Studies conducted in any healthcare setting or in the homes of older adults were eligible for inclusion. We included randomised and non-randomised controlled trials, controlled before-after studies, and interrupted time-series studies. Detailed eligibility criteria are presented in Table 1.

### Information sources and search strategy

We searched the following databases from the earliest available date through January 2025: MEDLINE (Ovid), EMBASE (Ovid), CINAHL (EBSCOhost), The Cochrane Library, Web of Science, and AMED (Ovid). Supplemental

**Table 1. Study eligibility criteria.**

|  | Inclusion criteria | Exclusion criteria |
|---|---|---|
| Participants | Healthcare practitioners (nurses, physiotherapists, general practitioners, occupational therapists, nursing assistants, pharmacists, physicians, primary care providers, paramedics) | Studies of CDS used by students. |
| Intervention | CDS used by healthcare practitioners in fall prevention interventions. We included both computerised and non-computerised CDS interventions. | Studies where CDS was not part of the intervention in at least one study group or arm. |
| Comparison | Usual care or no treatment. | |
| Outcomes | We included studies that reported at least one primary or secondary outcome.<br>Primary:<br>• Healthcare practitioner performance: Adherence to recommended practice and adherence to recommended medication prescribing and review.<br>• Patient outcomes: Rate of falls (i.e., number of falls per unit of follow-up time); risk of falling (i.e., the number of older adults who had one or more falls); and fall injuries.<br>Secondary:<br>• Deaths and hospitalisations. | Studies that did not report at least one primary or secondary outcome. |
| Design | RCTs: IRPGT and CRT; NRCTs; CBA studies; and ITS studies [42]. | Uncontrolled before-after, cross-sectional, case-control, and cohort studies; as well as protocols, editorials, opinion papers, and conference abstracts.<br>Studies without a control group, except for ITS studies with multiple data points collected before and after the implementation of the intervention. |
| Setting | Any healthcare setting or in the homes of older adults. | Studies conducted in a setting other than a healthcare institution or in the homes of older adults. |

CDS: Clinical Decision Support; RCT: Randomised controlled trial; IRPGT: Individually-randomised parallel-group trial; CRT: Cluster-randomised trial; NRCT: Non-randomised controlled trial; CBA: Controlled before-after study; ITS: Interrupted time-series.

searches were performed in Google Scholar (S3 Table). The literature search strategy was developed in cooperation with two librarians from the Literature Search Resource/Expert Group at Oslo Metropolitan University (OsloMet). The searches were developed to be highly sensitive and included terms synonymous with the fundamental concepts underlying CDS (e.g., decision rules, reminder systems, algorithms), falls (e.g., accidental falls, slip), and fall prevention (e.g., accident prevention, safety management). We did not impose any language or study design restrictions on the literature searches. To maximise the sensitivity of the search and capture a broader range of potentially relevant studies, study design was not restricted in the search strategy. One reviewer (RS) used the advanced search feature in the Physiotherapy Evidence Database (PEDro) to conduct manual searches for relevant clinical trials. Additionally, we searched the reference lists of the included articles and relevant systematic reviews. Cited reference searching for all included articles was carried out in Web of Science. See S3 Table for the complete search strategy.

## Selection process

Search results were imported into EndNote, where duplicates were removed. Next, all unique abstracts and full text articles were uploaded to the Covidence systematic review software [44]. Two reviewers (RS and either TB, NRO, or LAHK) independently screened titles, abstracts, and full texts against the eligibility criteria. Any disagreements were resolved through consensus or, if needed, by the decision of a third reviewer. Consensus between reviewers was required at both the title and abstract screening stage, as well as during the full-text screening stage.

## Data collection process

We used an adapted version of the EPOC data collection form to extract relevant data from the included studies. The data extraction form was piloted on three reports by two reviewers. One reviewer (RS) independently extracted the data, and another reviewer (TB) checked the extracted data against five arbitrarily selected papers. Any disagreements between reviewers were resolved by consensus, with consultation from a third reviewer if necessary. To address missing outcome data, we contacted the corresponding authors of eight included reports via email. If no responses were received, follow-up emails were sent after two weeks, with a maximum of three email attempts per author. Our efforts to obtain missing data were successful in four instances. All data used in the analyses were obtained directly from primary sources or through successful correspondence with authors, with no data imputation performed.

## Data items

**Outcomes.** For healthcare practitioner performance, we sought data on adherence to recommended practices, including the provision of specific advice, delivery of specific interventions, and adherence to referral guidelines [42]. We also collected data on medication outcomes, including medication review [45] and prescribing. For patient outcomes, we included fall risk, fall rate, and fall injury rate. Fall risk refers to the proportion of individuals who experienced at least one fall over a specific period, representing the likelihood of falling at the individual level. Fall rate and fall injury rate refer to the total number of falls or fall injuries per unit of time, e.g., per 1,000 person-years, accounting for multiple falls per individual. Fall injuries included both minor injuries, such as bruises or abrasions, and serious injuries, such as fractures or those requiring medical attention [46]. We also sought data for deaths and hospitalisations. If multiple results were reported in a study, we prioritised the primary outcome as specified by the authors or the outcome used for sample size calculation. If primary outcomes were not clearly identified, we prioritised results deemed to be most relevant to the research question, those indicating drug overuse rather than misuse or underuse (due to the significant fall risk imposed by polypharmacy [47]), and results based on whole-group analyses rather than subgroup analyses.

**Study characteristics.** We extracted the following data on study characteristics: authors, publication year, funding, country, design, setting, duration, healthcare practitioners delivering the intervention (type and number), and patients receiving the intervention (number, sex, age).

**Interventions and CDS.** We extracted the characteristics of experimental and control interventions using the Template for Intervention Description and Replication (TIDieR) checklist [48], and using domains 1 and 3 from the intervention complexity assessment tool for systematic reviews (iCAT_SR) [49]. Interventions were categorised into 1) manual fall risk assessment and interventions based on CDS; 2) medication review and recommendations made to physician; 3) automatically generated fall risk based on prediction models and recommended interventions; 4) guided medication dosing using computerised CDS; and 5) computerised CDS presented to paramedics on hand-held tablets [50,51]. Design factors of CDS tools were extracted using the GUideline Implementation with DEcision Support (GUIDES) framework [51] and elements from the two-stream model [52].

## Study risk of bias assessment

Two review authors independently assessed risk of bias (RoB) for outcomes indicating healthcare practitioner performance, fall rates, fall risk, and fall injuries. We used the Revised Cochrane Risk of Bias tool for randomised trials (RoB 2.0) [53] for included RCTs, and the Risk Of Bias In Non-randomised Studies of Interventions (ROBINS-I) tool [54,55] for included non-randomised studies. Any disagreements between reviewers were resolved by consensus, and if necessary, a third reviewer was consulted.

## Synthesis methods and effect measures

**Healthcare practitioner performance.** We performed two analyses related to healthcare practitioner performance, both of which used vote-counting based on direction of effect [41,56]. In the analysis of adherence to recommended

practice, we included outcomes indicating the use of recommended intervention components or adherence to the intervention protocol or to the referral guideline [42]. In the analysis of medication outcomes, we included outcomes indicating changes in the use of fall-risk-increasing drugs (FRIDs), polypharmacy, drug underuse, or drug-related problems [31]. To decide which results were eligible for the analyses, we tabulated each study result and compared it against the eligibility criteria. Each effect estimate was dichotomised into 'favouring CDS' or 'favouring control', based on the observed direction of effect alone. We used the sign test for differences in proportions and presented the proportion of results favouring CDS along with 95% confidence intervals (CIs) using the Wilson interval method [57]. The data used for analyses of healthcare practitioner performance outcomes are available in S1 Appendix.

**Patient outcomes.** Data on fall risk, fall rate, and fall injury rate were pooled into meta-analyses in line with previous work [9,58]. We assumed that underlying study effects followed a normal distribution and used random effects models and restricted maximum likelihood (REML) estimation methods [59]. Stata, version 18.0, StataCorp, College Station, Texas [60], was used for all analyses and pooled results were presented in forest plots. Results were presented as odds ratios (OR) for fall risk, and as rate ratios (RaR) for fall rate and fall injury rate, along with 95% CIs. Variation between study results (heterogeneity) was assessed by means of a visual inspection of forest plots and the $I^2$ statistic [59]. If $I^2$ values were 50% or higher, we sought potential explanations for the heterogeneity using post hoc subgroup analyses by RoB (low or some concerns for RoB versus high or serious RoB), study setting (community-dwelling versus hospital or residential care), and patient age (mean age of < 80 years versus ≥ 80 years). If point estimates and confidence intervals were missing but other information was available, e.g., number of events and the follow-up time in both comparison groups, we used established formulas to estimate the point estimates and confidence intervals [61]. Only one outcome per study was included in the analyses to ensure independence between studies. The data used for meta-analyses are available in S2 Appendix. Control intervention fall risks, fall rates, and fall injury rates were derived from a US report on the epidemiology of falls among older adults [62], due to the limited reporting of absolute numbers in the studies included in this review. To report the absolute effects on fall risk, odds ratios were first converted to relative risks using the formula provided in appendix 3 of the Core GRADE 2 article [63].

### Publication bias assessment

The possibility of publication bias was assessed through inspection of funnel plots of effect estimates against their standard errors for analyses that contained at least 10 effect estimates [41]. The Egger test for small-study effects and the trim-and-fill analysis were used for quantitative assessment of publication bias [64].

### Assessment of certainty of the evidence

We judged the certainty of the evidence using the Grading of Recommendations Assessment, Development and Evaluation (GRADE) approach [63,65,66], applied to each outcome from vote-counting analyses and meta-analyses. Despite extensive literature searches conducted with the assistance of two librarians at OsloMet, no minimal important difference (MID) values were identified for fall risk, fall rate, or fall injury rate. Certainty was therefore rated based on whether the true effect lies on the observed side of null, with the null effect used as the threshold. Relative and absolute effects were reported alongside plain-language statements, following recommendations from GRADE guidance [67,68].

## Results

### Study selection

Fig 1 presents an overview of the study selection process. The primary database searches identified 6,572 unique records, of which 6,527 were excluded after title and abstract screening. A total of 28 publications describing 25 unique studies were selected for inclusion [23–28,38,39,69–88]. S4 Table provides a summary of the excluded studies and the reasons for their exclusion.

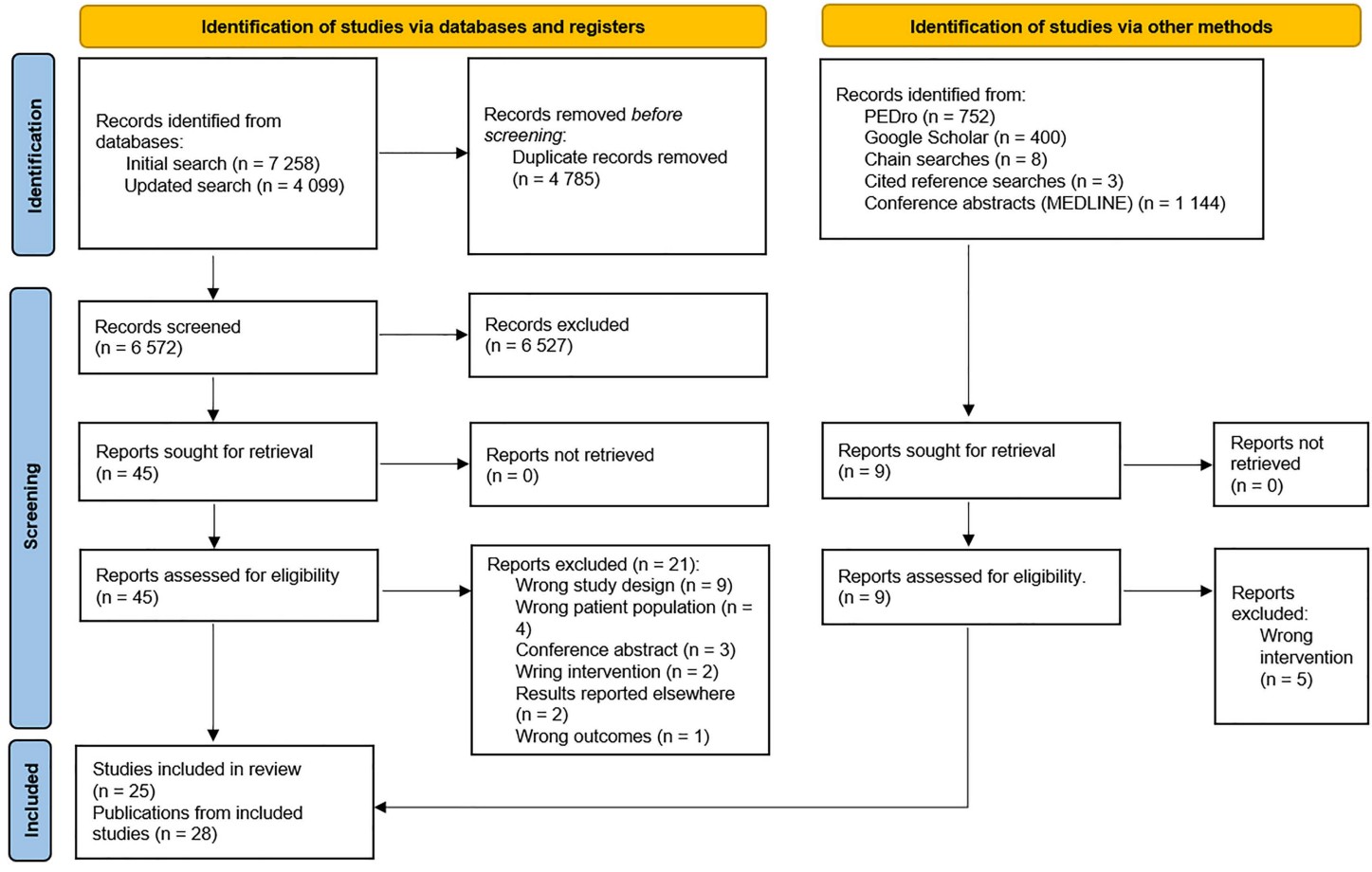

**Fig 1. PRISMA flow diagram.**

## Study characteristics

**Study design, participants, and settings.** Table 2 presents an overview of the characteristics of the included studies. The 25 studies comprised 13 cluster-randomised trials (CRTs), seven individually randomised parallel group trials (IRPGTs), four non-randomised controlled trials (NRCTs), and one controlled before-after (CBA) study. The study duration was median (min–max) 12 (1–60) months. The interventions were primarily delivered by nurses (16 studies; 64%), physicians (15 studies; 60%), pharmacists (6 studies; 24%), and/or physiotherapists (4 studies; 16%) to a median of 1,433 patient participants, with sample sizes ranging from 312 to 46,245. Twelve studies (48%) had eligibility criteria related to an increased risk of falls, such as a history of previous falls or the use of fall-risk-increasing drugs. Most studies were conducted in hospitals (44%) or primary care practices (36%) and took place in North America (52%) or Europe (28%).

**Interventions and CDS tools.** Table 3 presents an overview of the characteristics of the interventions and CDS tools assessed in the included studies. Most interventions (15 studies; 60%) delivered CDS to healthcare practitioners to aid with fall risk assessments, followed by an offer to the patient of specific fall prevention interventions. The majority of interventions (22 studies; 88%) consisted of more than one component delivered as a bundle, i.e., there was a defined order in the delivery of the components, such as conducting a medication review prior to implementing medication changes. Interventions were primarily directed at either one type of healthcare practitioner (12 studies; 48%) or at two or more types of healthcare practitioners within the same healthcare setting (11 studies; 44%). See S5 Table for a

**Table 2. Characteristics of included studies.**

| Author & year | Country | Design | Setting | Study duration (months) | Healthcare practitioners | Patients | Outcomes |
|---|---|---|---|---|---|---|---|
| Aizen 2015 | Israel | CRT | Rehabilitation geriatric hospital | 6 | Nurses n = ? | Patients admitted to hospital n = 508 (200 in CDS and 308 in CG), 52.2% female, mean 84.3 years | -Rate of falls |
| Barker 2016 | Australia | CRT | Hospital | 12 | Nurses n = ? | Patients admitted to hospital n = 35,264 (17,698 in CDS and 17,566 in CG), 49.5% female, median 67.5 years | -Adherence to recommended practice -Rate of falls -Rate of fall injuries |
| Bhasin 2020 | USA | CRT | Primary care practices | 44 | Nurses n = ? | Community-dwelling at increased risk for fall injuries n = 5,451 (2,802 in CDS and 2,649 in CG), 62% female, mean 79.7 years | -Rate of fall injuries |
| Blalock 2020 | USA | CRT | Community pharmacies | 24 | Pharmacists n = ? | Adults ≥65 years using either four or more chronic medications or ≥1 medication associated with increased fall risk, n = 3,213 (1,467 in CDS and 1,745 in CG) | -Medication outcomes -Risk of falling |
| Blum 2021 | Switzerland, Netherlands, Belgium, Republic of Ireland | CRT | Hospitals | 12 | Physicians and pharmacists n = ? | Adults years with multimorbidity (≥3 chronic conditions) and polypharmacy (≥5 drugs used long term) n = 2,008 (963 in CDS and 1,045 in CG), 44.7% female, median 79 years | -Rate of falls -Medication outcomes -Mortality -Hospital admissions |
| Byrne 2005 | USA | CBA | Nursing homes | 33 | Nurses n = 153 | n = ?,? females, Mean age in IG: 82.5 (group 1) and 80.9 (group 2); in CG: 79.2 (group 5) and 82.2 (group 6) | - Rate of falls |
| Carroll 2012 | USA | CRT | Urban hospitals (academic medical centres & community hospitals) | 6 | Nurses n = ? | Patients admitted during study period n = 364,? female,? age | -Adherence to recommended practice |
| Dykes 2010 | | | | | | Patients admitted during study period n = 10,264 (5,160 in CDS and 5,104 in CG), 54.5% female, mean 78.8 years (among patients 65 years or older) | -Rate of falls -Rate of fall injuries -Adherence to recommended practice |
| Clemson 2024 | Australia | CRT | Primary care practices | 12 | Physicians n = 75 Allied health professionals n = 342 | Community-dwelling older adults who had had a fall in the past year or were concerned about falling n = 560 (275 in CDS and 285 in CG), 67.9% female, mean 78.6 years | -Rate of falls -Adherence to recommended practice |
| Dykes 2020 | USA | NRCT | Hospital (academic medical centres) | 42 | Nurses n = ? | Patients admitted during study period n = 37,231 (17,948 in pre-intervention group and 19,283 in post-intervention group), 53.8% female, mean 60.8 years | -Rate of falls -Rate of fall injuries |
| Elley 2008 | New Zealand | IRPGT | Primary care practices | 12 | Nurses n = 54 | Adults who had fallen in the past 12 months n = 312 (155 in CDS and 157 in CG), 68.9% female, mean 80.8 years | -Rate of falls |
| Ferrer 2014 | Spain | IRPGT | Primary care practices | 24 | Physicians and nurses n = ? | Community-dwelling older adults born in 1924 (85 years of age at study start) n = 328 (164 in CDS and 164 in CG), 61.1% female | -Risk of falling -Time to first, second, and recurrent falls -Hospital admissions |
| Frankenthal 2014 | Israel | IRPGT | Hospital (Chronic care geriatric facility) | 12 | Pharmacist n = ? | Residents prescribed at least one medication n = 359 (183 in CDS and 176 in CG), 66.6% female, mean 82.7 years | -Rate of falls -Hospital admissions |

*(Continued)*

| Author & year | Country | Design | Setting | Study duration (months) | Healthcare practitioners | Patients | Outcomes |
|---|---|---|---|---|---|---|---|
| Gallagher 2011 | Ireland | IRPGT | University hospital | 6 | Physicians n = ? | Patients admitted via the emergency department under care of a GP n = 382 (190 in CDS and 192 in CG), 53.1% female, median 74.5 years (IG) and 77 (CG) | -Medication outcomes -Risk of falling -Hospital admissions |
| Ganz 2015 | USA | NRCT | Primary care practices | 24 | Physicians, nurse practitioner, physician assistant n = 44 | Patients who screened positive for fall risk n = 1,791 (1,187 in CDS and 604 in control), 72% female, mean 82.9 years | -Rate of fall injuries |
| Wenger 2010 | | | | 12 | | Patients who screened positive for falls or fear of falling and UI: n = 1,211 (586 in CDS and 625 in CG), 71.7% female, mean 83 years | -Adherence to recommended practice |
| Ganz 2022 | USA | CRT | Primary care practices | 60 | Nurses n = ? | Community-living persons at increased risk for serious fall injuries n = 5,451 (2,802 in CDS and 2,649 in CG), 62% female, mean 79.7 years | -Rate of falls -Rate of falls leading to medical attention -Hospital admissions |
| Groshaus 2012 | Canada | NRCT | Acute care hospitals | 3 | Nurses n = ? | Patients residing on study units n = ?,? female,? years | -Adherence to recommended practice -Risk of falling |
| Healey 2004 | UK | CRT | District general hospital | 12 | Nurses n = ? | All older adults who received care in the wards during the study period n = 1,654 (905 in CDS and 749 in CG), 60% female, mean 81.3 years | -Rate of falls -Rate of fall injuries |
| Lightbody 2002 | UK | IRPGT | University hospital | 6 | Nurses n = ? | Older adults discharged from Accident and Emergency Department after a fall n = 348 (171 in CDS and 177 in CG), 74.4% female, median 75 years | -Medication outcomes -Risk of falling -Rate of falls -Hospital admissions |
| Logan 2021 | UK | CRT | Long-term care homes | 12 | Care home staff n = 3,609 | Long-term care home residents in care homes for older adults n = 1,657 (775 in CDS and 882 in CG), 67.9% female, mean 85 years | -Rate of falls -Mortality |
| Mahoney 2007 | USA | IRPGT | Home visits | 12 | Registered nurse, physical therapist, physician n = ? | Community-dwelling adults with two falls in the past year or one fall in the previous two years with injury or balance problems n = 349 (174 in CDS and 175 in CG), 78.5% female, mean 80 years | -Rate of falls -Hospital admissions |
| Peterson 2007 | USA | IRPGT | Tertiary care hospital | 9 | Physicians n = 778 | Inpatients ≥ receiving care on one of the order entry wards (i.e., emergency room, intensive care units, subacute units) n = 2,981,? female, = years | -Adherence to recommended practice |
| Phelan 2024 | USA | CRT | Primary care practices | 18 | Physicians | Community-dwelling adults aged ≥ 60 years, prescribed at least 1 medication from any of 5 targeted medication classes (opioids, sedative-hypnotics, skeletal muscle relaxants, tricyclic antidepressants, and first-generation antihistamines for at least 3 consecutive months n = 2,367 (1,106 in CDS and 1,261 in CG), 63% female, mean 70.6 | -Time to first medically treated fall (risk of fall injuries) -Medication outcomes |

*(Continued)*

**Table 2.** (Continued)

| Author & year | Country | Design | Setting | Study duration (months) | Healthcare practitioners | Patients | Outcomes |
|---|---|---|---|---|---|---|---|
| Snooks 2014 | UK | CRT | Emergency ambulance services | 1 | Paramedics n=42 | Community-dwelling older adults living in the catchment area of a participating falls service n=779 (436 in CDS and 343 in CG), 63.4% female, median 82.5 years | -Adherence to recommended practice -Risk of falling -Mortality -Hospital admissions |
| Tamblyn 2012 | Canada | CRT | Primary care practices | 23 | GPs n=81 | Patients with a prescription for a psychotropic drug n=5,628 (2,887 in CDS and 2,741 in CG), 67.1% female, mean 75.2 years | -Medication outcomes |
| Weber 2008 | USA | CRT | Primary care practices | 15 | Pharmacists and GPs n=? | Community-dwelling patients at risk for falls n=620 (413 in CDS and 207 in CG), 79.2% female, mean 76.9 years | -Risk of falling -Medication outcomes |
| Wenger 2009 | USA | NRCT | Primary care practices | 32 | Physicians n=40 | Community-dwelling patients who had at least one of three geriatric conditions: falls and gait impairment, urinary incontinence, and cognitive impairment n=644 (357 in CDS and 287 in CG), 66% female, mean 81 years | -Adherence to recommended practice |

CBA: Controlled Before-After study; CCDS: Computerised Clinical Decision Support; CDS: Clinical Decision Support group; CG: Control group; CRT: Cluster-Randomised Trial; ED: Emergency Department; GP: General Practitioner; IG: Intervention Group; ITS: Interrupted Time-Series study; IRPGT: Individually-Randomised Parallel-Group Trial; NRCT: Non-Randomised Controlled Trial; UI: Urinary Incontinence.

detailed description of the interventions. Most CDS tools were designed as algorithms presented as if-then statements or as checklists with risk factors for falls, followed by recommendations for fall prevention interventions that were either generated automatically or chosen manually from a list (68%). Regarding delivery methods, 15 CDS tools (60%) were electronic, three (12%) were delivered both electronically and on paper, two (8%) were paper-based, and for five studies (20%), the delivery method was unclear. Among the electronic CDS tools, nine (36%) included alerts, reminders, or prompts, while the others delivered decision support on demand. Control interventions included usual care (76%), usual care plus a falls information pamphlet (4%), usual care plus an offer of two social visits (4%), home safety visits (4%), paper-based CDS (4%), and no treatment (4%). One study (4%) failed to mention control. See S6 Table for a detailed description of the CDS design factors.

**Risk of bias in included studies.** A detailed description of RoB assessments is available in S7 Table. We assessed a total of 40 results across five outcomes. Out of 33 results from randomised trials, 19 were judged to have high RoB, 10 had some concerns, and four were rated as having a low RoB. All seven results from non-randomised studies were judged to be at serious RoB, primarily owing to concerns for bias related to confounding.

## Results of individual studies

Results from each individual study are available in S8 Table.

## Healthcare practitioner performance

**Adherence to recommended practice.** Five CRTs [24,25,28,69,86] and three NRCTs [78,80,88] reported results on adherence to recommended practice regarding fall prevention (Fig 2). Outcomes across these studies included fall

**Table 3. Characteristics of interventions and Clinical Decision Support.**

| Study | Type of intervention | Active components | n types of intervention deliverers | Organisational levels | CDS format | CDS features |
|---|---|---|---|---|---|---|
| Aizen 2015 | 1 | High | 1: Nurses | Low | Electronic | Risk assessment tool with intervention recommendations based on risk |
| Barker 2016 | 1 | High | 3: Nurse, site clinical leader, champion | Intermediate | – | Checklist with risk factors and recommended interventions; Reminders on the use of intervention components |
| Bhasin 2020, Ganz 2022 | 1 | High | 3: Nurse, primary care physician, pharmacist | Intermediate | Electronic | Algorithm with risk factors and intervention recommendations |
| Blalock 2020 | 2 | High | 2: Pharmacist, primary care physician | Intermediate | Electronic | Adapted STEADI algorithm with risk factors and intervention recommendations |
| Blum 2021 | 2 | High | 3: Pharmacist, hospital physician, general practitioner | High | Electronic | Lists of FRIDs (STOPP/START criteria) |
| Byrne 2005 | 3 | High | 1: Nurse | Low | Electronic | Automatically generated fall risk estimates and intervention recommendations |
| Carroll 2012, Dykes 2010 | 1 | High | 1: Nurse | Low | Electronic and paper-based | Alerts printed on paper to hang over bed |
| Clemson 2024 | 1 | High | > 7: Primary care physicians, physiotherapists, occupational therapists, nurses, podiatrists, pharmacists, exercise physiologists, and other professions | High | Electronic and paper-based | iSOLVE algorithm; Fall risk assessment checklist; GP fall risk assessment chart; Tailoring interventions to fall risk chart; Risk information automatically sent to GP; Case studies which illustrate the algorithm and tailoring options; Examples of how to talk with patients about falls |
| Dykes 2020 | 1 | High | 1: Nurse | Low | Electronic | Checklist with risk factors and recommended interventions |
| Elley 2008 | 1 | High | 3: Nurse, trained practitioner, physiotherapist | Intermediate | – | Algorithm with risk factors and intervention recommendations |
| Ferrer 2014 | 1 | High | 2: Nurse, primary care physician | Intermediate | – | Algorithm with risk factors and intervention recommendations |
| Frankenthal 2014 | 2 | High | 2: Pharmacist, hospital physician | Intermediate | Electronic | Lists of FRIDs (STOPP/START criteria) |
| Gallagher 2011 | 2 | High | 1: Hospital physician | Low | – | Lists of FRIDs (STOPP/START criteria) |
| Ganz 2015, Wenger 2010 | 1 | High | 3: Primary care physician, nurse, physician assistant | Intermediate | Electronic | Medical record prompts |
| Groshaus 2012 | 1 | High | 1: Nurse | Low | Electronic | Electronic order set |
| Healey 2004 | 1 | High | 1: Nurse | Low | Paper-based | Checklist with risk factors and intervention recommendations |
| Lightbody 2002 | 1 | High | 1: Nurse | Low | – | Checklist with risk factors and intervention recommendations |
| Logan 2021 | 1 | High | 3: Nurse, physiotherapist, occupational therapist | Intermediate | Paper-based | Checklist with risk factors and intervention recommendations |
| Mahoney 2007 | 1 | High | 3: Nurse, physiotherapist, primary care physician | Intermediate | Electronic | Algorithm with automatically generated intervention recommendations |
| Peterson 2007 | 4 | Low | 1: Hospital physician | Low | Electronic | Guided medication dosing; Prompts presented on-screen |

*(Continued)*

**Table 3.** (Continued)

| Study | Type of intervention | Active components | n types of intervention deliverers | Organisational levels | CDS format | CDS features |
|---|---|---|---|---|---|---|
| Phelan 2024 | 2 | Intermediate | 1: Primary care physician | Low | Electronic | Evidence-based pharmaceutical opinions; Deprescribing pearls with conversation starters |
| Snooks 2014 | 5 | High | 1: Paramedic | Low | Electronic | Prompts to start assessment; Algorithm with automatically generated care plan |
| Tamblyn 2012 | 4 | Low | 1: Primary care physician | Low | Electronic | Predictive model to automatically estimate risk of injury; Alerts when patient was prescribed a FRID; Graphics presenting risk estimates; Guided medication dosing |
| Weber 2008 | 2 | High | 3: Pharmacist, geriatrician, primary care physician | Intermediate | Electronic | Guided medication dosing; Alerts with patient's fall risk |
| Wenger 2009 | 1 | High | 3: Nurse, primary care physician, medical assistant | Intermediate | Electronic and paper-based | Medical record prompts with suggestions for appropriate action |

CDS: Clinical Decision Support; FRIDs: Fall-risk-increasing drugs; GP: General practitioner; STEADI: Stopping Elderly Accidents, DEaths, and Injuries; STOPP/START: Screening Tool of Older Person's Prescriptions and Screening Tool to Alert doctors to Right Treatment.

**Type of intervention:**

1: Fall risk assessment and interventions based on CDS.

2: Medication review and recommendations made to physician.

3: Automatically generated fall risk based on prediction models, followed by recommended interventions.

4: Guided medication dosing using computerised CDS.

5: Computerised CDS presented to paramedics on hand-held tablets.

**Active components (iCAT_SR domain 2):**

High: More than one component and delivered as a bundle (clear order in the delivery of the components) (high level of complexity).

Intermediate: More than one component and delivered as a package (no specific order) (intermediate level of complexity).

Low: One component (low level of complexity).

**Organisational levels and categories targeted by the intervention (iCAT_SR domain 3):**

High: Intervention directed at two or more healthcare settings, e.g., primary care and hospitals (multi-level).

Intermediate: Intervention directed at two or more categories of healthcare practitioners within the same healthcare setting, e.g., nurse and physiotherapist in primary care (multi-category).

Low: Intervention directed at one category of healthcare practitioner, e.g., nurses (single category).

risk documentation and provision of recommended intervention components. The median follow-up time was 7 months (range: 1–13 months). All eight results favoured the intervention (100%; 95% CI: 68% to 100%; p < 0.01). The certainty of evidence was low due to risk of bias (Table 4). Overall, these findings suggest that CDS may improve healthcare practitioners' adherence to recommended practice in fall prevention.

**Adherence to recommended medication review and prescribing.** Five CRTs [23,39,70,72,87] and four IRPGTs [75,76,82,85] reported indicators of medication outcomes (Fig 2). The common outcome across these studies was the reviewing and prescribing of drugs that may increase fall risk. The median follow-up time was 9 months (range: 0–23 months). All nine results favoured the intervention (100%; 95% CI: 70% to 100%; p < 0.01). The certainty of evidence was moderate due to risk of bias. These findings suggest that CDS likely improves medication reviewing and prescribing outcomes in the context of fall prevention.

## Patient outcomes

**Fall risk.** Nineteen of the included studies reported outcome data on fall risk (proportion of participants who fell) and/or fall rate (number of falls per x person-years of follow-up), enabling meta-analyses of these outcomes. Ten

| Study | Direction of effect | | Results | Specific outcome | RoB |
|---|---|---|---|---|---|
| | Favours CDS | Favours control | | | |
| **Adherence to recommended practice** | | | | | |
| Barker (2016) | X | | Rate ratio 3.05 (95% CI 2.14, 4.34) | Use of all 6-PACK programme components (fall risk tool and six interventions) during eight months FU | ! (yellow) |
| Carroll (2012) | X | | 89% in IG, 64% in CG (p < 0.0001) | Proportion of patient records reviewed that had fall risk documented on the plan of care during six months FU | − (red) |
| Clemson (2024) | X | | 0.90 units (95% CI 0.33, 1.46) | Changes in GPs' engagement in fall prevention activities at 12 months FU, including risk assessments, medication reviews, and providing advice, compared to the control group. Composite score of how often they engaged with older patients in managing falls, max 9 | − (red) |
| Dykes (2010) | X | | 94% in IG, 81% in CG | Adherence to intervention protocol during six months FU (Morse Falls Scale completion) | − (red) |
| Groshaus (2012) | X | | Mean difference 3.1 (95% CI 1.9, 5.3) | Mean rate of use of the order set at the end of each two-week period for a total of three months | −* (orange) |
| Wenger (2009) | X | | 44% in IG, 23% in CG (p < 0.001) | Proportion of recommended care (quality indicators) for falls provided to patients during 13 months FU | −* (orange) |
| Wenger (2010) | X | | 60% in IG, 37.6% in CG (p < 0.001) | Proportion of recommended care (quality indicators) for falls provided to patients during 12 months FU | −* (orange) |
| Snooks (2014) | X | | Odds ratio 2.04 (95% CI 1.12, 3.72) (p = 0.021) | Odds of a patient being referred to a falls service during one month FU in intervention group compared with control group | ! (yellow) |
| | Sign test p < 0.01 | | | | |
| **Medication outcomes** | | | | | |
| Blalock (2020) | X | | Mean difference of −0.12 in intervention group compared with −0.08 in control group (p = 0.05) | Change in use of fall-risk-increasing drugs in individuals who screened positive for fall risk (DBI score) from 12-month pre-intervention period to 12-month post-intervention period* | ! (yellow) |
| Blum (2021) | X | | Odds ratio 0.99 (95% CI 0.82, 1.20) | Presence of drug overuse based on STOPP criteria during two months FU in intervention group compared with control group | ! (yellow) |
| Frankenthal (2014) | X | | Mean (SD) 7.3 (2.7) in IG, 8.9 (3.2) in CG (p < 0.001) | Number of medications prescribed at 12-month FU | ! (yellow) |
| Gallagher (2011) | X | | Absolute risk reduction 35.7% (95% CI 26.3%, 44.9%) | Unnecessary polypharmacy, the use of drugs at incorrect doses, and potential drug-drug-diseases and drug-disease interactions at discharge | − (red) |
| Lightbody (2002) | | X | 608 in IG, 684 in CG (p = 0.41) | Number of daily medications at six-month FU | − (red) |
| Peterson (2007) | X | | Median (IQR) 2.5 (1.0, 4.0) in IG, 3.0 (1.5, 5.0) in CG (p < 0.001) | Ratio between prescribed and recommended medication dose during nine months FU | − (red) |
| Phelan (2024) | X | | Adjusted relative risk 1.24 (95% CI 0.90, 1.70) | Discontinuation of medications (defined as no prescription fill for 90 days), summarized across all target medication classes, referred to as "first target medication", at six months FU | − (red) |
| Tamblyn (2012) | X | | Mean reduction 1.7 (95% CI 0.2, 3.2) | Reduction in risk of injury (based on psychotropic medications and non-modifiable risk factors) at 23-month FU | + (green) |
| Weber (2008) | X | | Mean reduction 0.496 (p = 0.088) | Change in number of active medications during 12 months FU | − (red) |
| | Sign test p < 0.01 | | | | |

**Fig 2. Healthcare practitioner performance results.** FU: Follow-up; IQR: Interquartile range; DBI: Drug Burden Index; IG: Intervention group; CG: Control group; RoB: Overall risk of bias judgement. † Based on subgroup analysis and therefore not randomised. * Non-randomised study. Serious risk of bias based on ROBINS-I. Sign test: H0: n results favouring CDS = n results favouring control. Ha: n results favouring CDS ≠ n results favouring control.

**Table 4. Summary of findings.**

**CDS interventions compared with usual care for HCPs in fall prevention among older adults**

**Population:** HCPs, including nurses, physiotherapists, general practitioners, occupational therapists, nursing assistants, pharmacists, physicians, primary care providers, and paramedics
**Settings:** Hospitals, residential care, primary care, and the homes of older adults
**Intervention:** CDS targeted at HCPs
**Comparison:** Usual care

| Outcomes No of participants (studies) | Relative effects (95% CI) | Anticipated absolute effects* (95% CI) | | | Certainty of evidence |
|---|---|---|---|---|---|
| | | In control | CDS interventions | Difference | |
| **Adherence to recommended practice: fall risk assessments and interventions** Follow-up: median 7 (1–13) months [a] (8) | [b] | [b] | [b] | [b] | ⊕⊕OO Low[c] Due to very serious risk of bias |
| **Adherence to recommended medication review and prescribing** Follow-up: median 9 (0–23) months [a] (9) | [b] | [b] | [b] | [b] | ⊕⊕⊕O Moderate[d] Due to risk of bias |
| **Fall risk** Follow-up: median 9 (1–24) months > 13,636[e] (10) | OR 0.93 (0.85 to 1.01) | 287 per 1,000 | 273 per 1,000 (255–290) | 14 fewer fallers per 1,000 (32 fewer to 3 more) | ⊕⊕OO Low[f] Due to risk of bias and imprecision |
| **Rate of falls** | | | | | |
| In hospitals or residential care Follow-up: median 6 (3–21) months >50,054[e] (8) | RaR 0.74 (0.63 to 0.88) | 672 per 1,000 person-years | 497 per 1,000 person-years (423–591) | 175 fewer falls per 1,000 person-years (249 fewer to 81 fewer) | ⊕⊕⊕O Moderate[g] Due to risk of bias |
| In community-dwelling older adults Follow-up: median 12 (12–24) months 7,000 (5) | RaR 0.97 (0.93 to 1.00) | 672 per 1,000 person-years | 652 per 1,000 person-years (625–672) | 20 fewer falls per 1,000 person-years (47 fewer to 0 fewer) | ⊕⊕OO Low[h] Due to risk of bias and imprecision |
| In patients with mean age ≥ 80 years Follow-up: median 12 (3–12) months 6,264 (7) | RaR 0.72 (0.61 to 0.86) | 672 per 1,000 person-years | 484 per 1,000 person-years (410–578) | 188 fewer falls per 1,000 person-years (262 fewer to 94 fewer) | ⊕⊕⊕O Moderate[i] Due to risk of bias |
| In patients with mean age between 65 and 80 years Follow-up: median 12 (6–24) months 84,084 (6) | RaR 0.92 (0.84 to 1.01) | 672 per 1,000 person-years | 618 per 1,000 person-years (564–679) | 54 fewer falls per 1,000 person-years (108 fewer to 7 more) | ⊕⊕OO Low[j] Due to risk of bias and imprecision |
| **Rate of fall injuries** | | | | | |
| In patients with mean age between 65 and 80 years Follow-up: median 16.5 (6–24) months >50,979[e] (4) | RaR 0.80 (0.59 to 1.09) | 164 per 1,000 person-years | 131 per 1,000 person-years (97–179) | 33 fewer fall injuries per 1,000 person-years (67 fewer to 15 more) | ⊕⊕OO Low[k] Due to risk of bias and imprecision |
| In patients with mean age ≥ 80 years Follow-up: median 9 (6–12) months 3,445 (2) | RaR 1.29 (0.99 to 1.69) | 164 per 1,000 person-years | 212 per 1,000 person-years (162–277) | 48 more fall injuries per 1,000 person-years (2 fewer to 113 more) | ⊕OOO Very low[l] Due to very serious risk of bias and imprecision |

*Assuming a control group fall risk of 28.7%, a fall rate of 672 falls per 1,000 person-years, and a fall injury rate of 164 fall injuries per 1,000 person-years, based on data from Bergen et al. [62]. The risk in the intervention group is based on the assumed risk in the comparison group and the relative effect of the intervention (and its 95% CI).

CDS: clinical decision support; CI: confidence interval; HCPs: healthcare practitioners; OR: odds ratio; RaR: rate ratio.

**Explanations:**

[a]The numbers of participating healthcare practitioners were not reported.

[b]Not estimable due to the use of different and incompletely reported effect measures across studies.

[c]Very serious risk of bias due to confounding, deviations from intended interventions, and selection of reported results.

*(Continued)*

**Table 4.** (Continued)

[d]Risk of bias due to deviations from intended interventions.

[e]The number of participants were not reported in all studies.

[f]Risk of bias due to the randomisation process, deviations from intended interventions, and measurement of the outcome. Imprecision due to the upper bound of the 95% confidence interval crossing the null effect.

[g]Risk of bias in the randomisation process and due to deviations from intended interventions.

[h]Risk of bias due to deviations from intended interventions. Imprecision due to the upper bound of the 95% confidence interval including the null effect.

[i]Risk of bias due to deviations from intended interventions and selection of reported results.

[j]Risk of bias due to the randomisation process, confounding, and deviations from intended interventions. Imprecision due to the upper bound of the 95% confidence interval crossing the null effect.

[k]Risk of bias due to confounding and deviations from intended interventions. Imprecision due to the upper bound of the 95% confidence interval crossing the null effect.

[l]Very serious risk of bias due to the randomisation process and confounding. Imprecision due to the lower bound of the 95% confidence interval crossing the null effect.

studies were included in the meta-analysis on fall risk, with a median follow-up time of 9 months (range: 1–24 months) [23,27,72,74,76,79,80,82,86,87] (Fig 3). Of these studies, six were conducted in patients aged 65–80 years, three in patients aged 80 years or older, and one study did not report the age of patients. Furthermore, five studies were conducted in hospitals or residential care settings, while the other five focused on community-dwelling older adults. The overall estimated effect of CDS on fall risk was an odds ratio of 0.93 (95% CI: 0.85 to 1.01). Assuming a baseline fall risk of 28.7% [62], this 7% reduction in the odds of falling results in 14 fewer fallers per 1,000 older adults (95% CI: 32 fewer to 3 more) receiving the intervention compared with controls, during nine months of follow-up (Table 4).

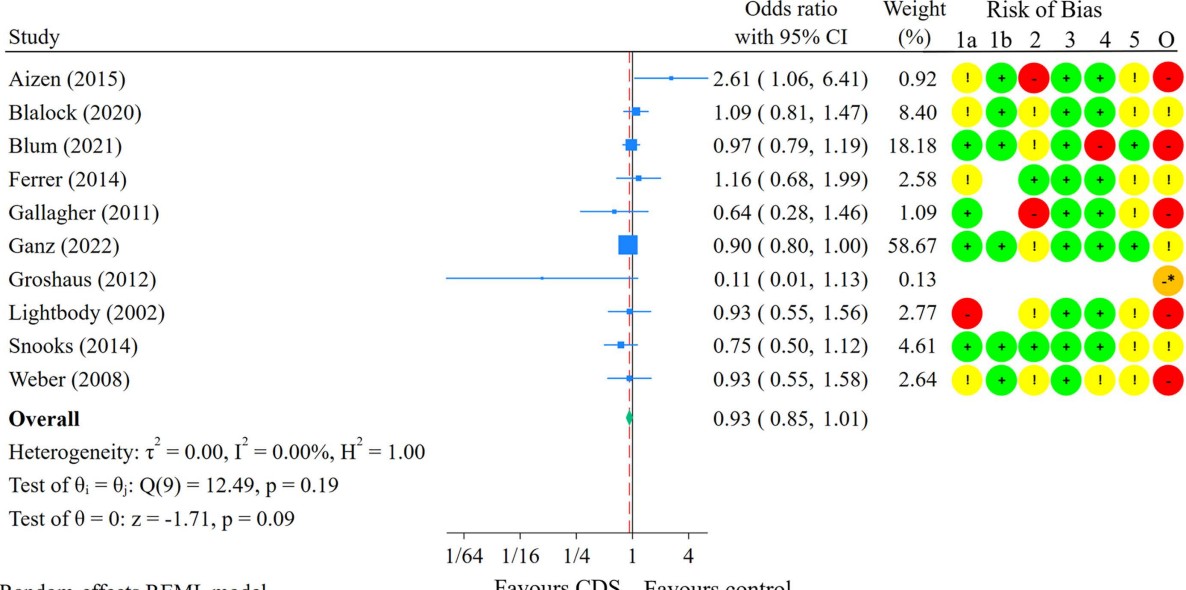

**Fig 3. Meta-analysis comparing CDS interventions with control on fall risk.** Red dashed line shows the point estimate for the meta-analysis overall. **Risk of Bias legend:** Circles: Green: Low risk of bias (RoB); Yellow: Some concerns for RoB; Red: High RoB; Orange: Serious RoB. 1a: Bias arising from the randomisation process. 1b: Bias arising from the timing of identification or recruitment of individual participants within clusters (for cluster-randomised only). 2: Bias due to deviations from intended interventions. 3: Bias due to missing outcome data. 4: Bias in measurement of the outcome. 5: Bias in selection of the reported result. O: Overall RoB judgement. * Non-randomised study. ROBINS-I was used to assess RoB.

Heterogeneity was unimportant ($I^2 = 0\%$; p = 0.19). Visual inspection of the funnel plot showed no clear sign of small-study effects (S1 Fig). The Egger test suggested no evidence of small-study effects (p = 0.89). Additionally, the trim-and-fill analysis imputed no studies, and therefore made no difference in the effect estimate (S2 Fig). The certainty of the evidence was low due to risk of bias and imprecision. While CDS aimed at healthcare practitioners may influence fall risk, the evidence does not demonstrate a statistically significant effect, and uncertainty remains regarding its effectiveness.

**Rate of falls.** Thirteen studies were included in the meta-analysis on fall rate [24,26–28,38,69,74,75,79,81–84] (Fig 4). Substantial heterogeneity was present in the effect estimate ($I^2 = 74\%$; p < 0.01). Subgroup analyses revealed statistically significant differences in fall rates when grouped by study setting (p < 0.01) and patients' age (p < 0.01). The results are therefore presented separately by study setting and patients' age.

Regarding subgroup analysis by study setting, the median follow-up time was 6 months (range: 3–21 months) for studies conducted in hospitals or residential care, and 12 months (range: 12–24 months) for studies on community-dwelling older adults. Among the eight studies conducted in hospitals or residential care, four included patients aged 80 years or older, and four included patients aged 65–80 years. Similarly, among the five studies focusing on community-dwelling older adults, three included patients aged 80 years or older, while two included patients aged 65–80 years. In hospitals or residential care, the overall estimated effect of CDS on fall rate was a rate ratio of 0.74 (95% CI: 0.63 to 0.88; $I^2 = 65\%$; p < 0.01). Assuming a baseline fall rate of 672 falls per 1,000 person-years [62], this 26% reduction in fall rate corresponds to 175 fewer falls per 1,000 person-years (95% CI: 249 fewer to 81 fewer) (Table 4). The certainty of the evidence was moderate due to risk of bias. These findings suggest that CDS aimed at healthcare practitioners likely reduces the rate of falls in hospitals and residential care settings.

For community-dwelling older adults, the overall estimated effect on fall rate was a rate ratio of 0.97 (95% CI: 0.93 to 1.00; $I^2 = 0\%$; p = 0.88). The certainty of the evidence was low due to risk of bias and imprecision. While CDS may influence fall rates in community-dwelling older adults, the effect is not statistically significant, and the certainty of the evidence is low.

Regarding the subgroup analysis by patients' age, the median follow-up time was 12 months (range: 3–12 months) for studies involving patients with a mean age ≥ 80 years, and 12 months (range: 6–24 months) for studies of patients with a mean age between 65 and 80 years (S3 Fig). Among the six studies including patients aged 65–80 years, four were conducted in hospitals or residential care settings, while two focused on community-dwelling older adults. Similarly, among the seven studies involving patients aged 80 years or older, four were conducted in hospitals or residential care, while three focused on community-dwelling older adults. The overall estimated effect of CDS on fall rate among patients with a mean age of ≥ 80 years was a rate ratio of 0.72 (95% CI: 0.61 to 0.86; $I^2 = 46\%$; p = 0.09). Assuming a baseline fall rate of 672 falls per 1,000 person-years [62], this 28% reduction in fall rate corresponds to 188 fewer falls per 1,000 person-years (95% CI: 262 fewer to 94 fewer). The certainty of the evidence was moderate due to risk of bias. These findings suggest that CDS likely reduces the rate of falls among patients with a mean age of ≥ 80 years.

For patients with a mean age between 65 and 80 years, the overall estimated effect on fall rate was a rate ratio of 0.92 (95% CI: 0.84 to 1.01; $I^2 = 31\%$; p = 0.09). The certainty of the evidence was low due to risk of bias and imprecision. While CDS may influence fall rates in patients aged 65–80 years, the effect is not statistically significant, and the certainty of the evidence is low. Visual inspection of the funnel plot showed no clear sign of small-study effects (S4 Fig). The Egger test suggested no evidence of small-study effects (p = 0.99). Additionally, the trim-and-fill analysis imputed no studies and therefore made no difference in the effect estimate (S5 Fig). There was no statistically significant subgroup difference in fall rates when stratified by RoB (p = 0.47) (S6 Fig).

**Rate of fall injuries.** Six studies were included in the meta-analysis on fall injury rate [24,26,28,77,79,81] (Fig 5). Substantial heterogeneity was present in the effect estimate ($I^2 = 71\%$, p = 0.03). Subgroup analyses revealed a statistically significant difference in fall injury rate between patients with a mean age of 65–80 years and patients with a mean age

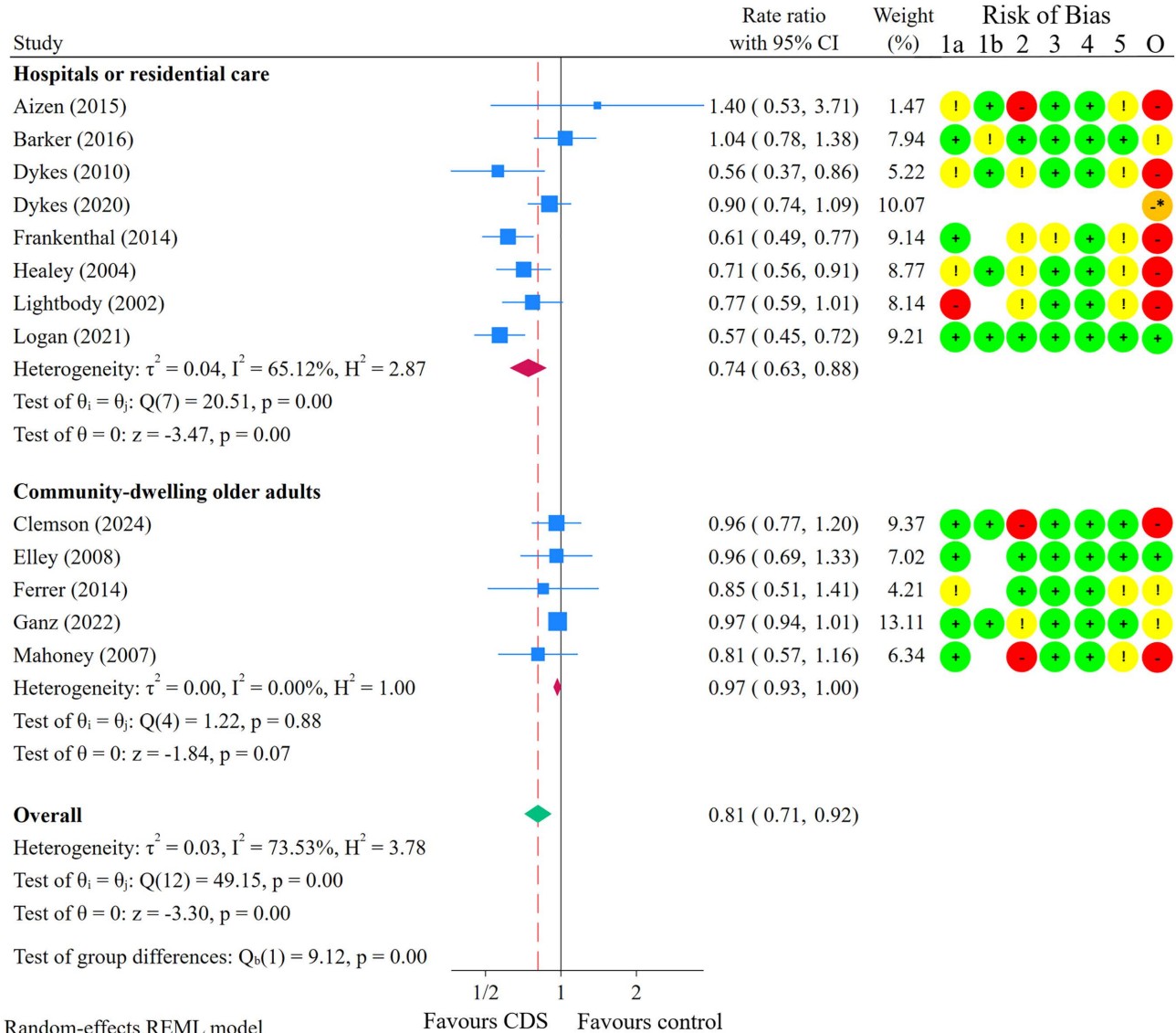

**Fig 4. Meta-analysis comparing CDS interventions with control on fall rate, subgroup analysis by study setting.** Red dashed line shows the point estimate for the meta-analysis overall. **Risk of Bias legend:** Circles: Green: Low risk of bias (RoB); Yellow: Some concerns for RoB; Red: High RoB; Orange: Serious RoB. 1a: Bias arising from the randomisation process. 1b: Bias arising from the timing of identification or recruitment of individual participants within clusters (for cluster-randomised only). 2: Bias due to deviations from intended interventions. 3: Bias due to missing outcome data. 4: Bias in measurement of the outcome. 5: Bias in selection of the reported result. O: Overall RoB judgement. * Non-randomised study. ROBINS-I was used to assess RoB.

of 80 years or older (p = 0.02). The results are therefore presented separately by patients' age. The median follow-up time was 16.5 months (range: 6–24 months) for studies including patients with a mean age of 65–80 years and 9 months (range: 6–12 months) for studies involving patients with a mean age of 80 years or older. The overall estimated effect of CDS in preventing fall injuries among patients aged 65–80 years was a rate ratio of 0.80 (95% CI: 0.59 to 1.09; $I^2 = 67\%$; p = 0.08), indicating a statistically non-significant result as the confidence interval crosses the null effect. Assuming a baseline fall injury rate of 164 fall injuries per 1,000 person-years [62], this 20% reduction in fall injury rate corresponds

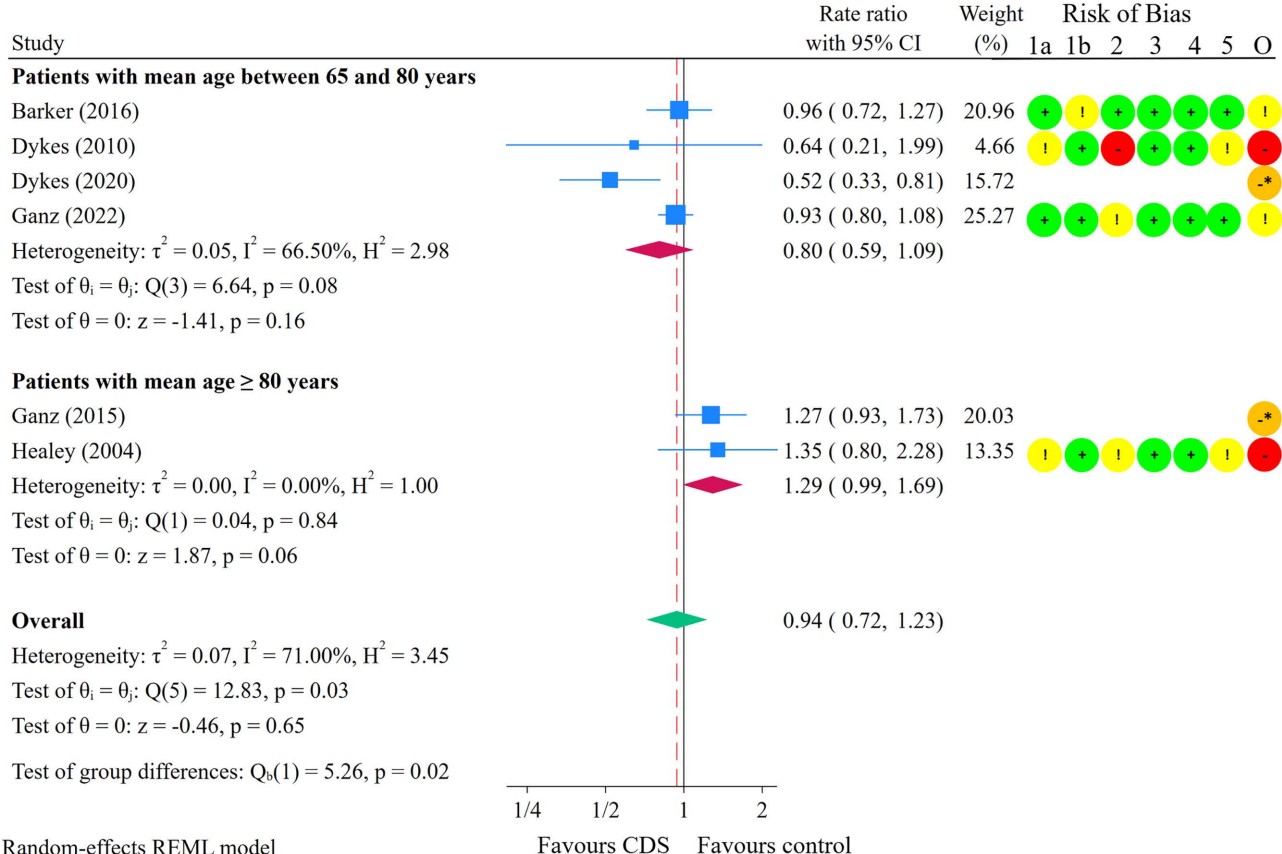

**Fig 5. Meta-analysis comparing CDS interventions with control on fall injury rate, subgroup analysis by patients' age.** Red dashed line shows the point estimate for the meta-analysis overall. Circles: Green: Low risk of bias (RoB); Yellow: Some concerns for RoB; Red: High RoB; Orange: Serious RoB. 1a: Bias arising from the randomisation process. 1b: Bias arising from the timing of identification or recruitment of individual participants within clusters (for cluster-randomised only). 2: Bias due to deviations from intended interventions. 3: Bias due to missing outcome data. 4: Bias in measurement of the outcome. 5: Bias in selection of the reported result. O: Overall RoB judgement. * Non-randomised study. ROBINS-I was used to assess RoB.

to 33 fewer fall injuries per 1,000 person-years (95% CI: 67 fewer to 15 more) in older adults receiving the intervention compared to controls (Table 4). The certainty of the evidence was low due to risk of bias and imprecision. While CDS may influence fall injury rates in patients aged 65–80 years, the effect is not statistically significant.

For patients with a mean age of 80 years or older, the overall estimated effect on fall injury rate was a rate ratio of 1.29 (95% CI: 0.99 to 1.69; I²=0%; p=0.84). The certainty of the evidence was very low due to risk of bias and imprecision. We are very uncertain about the effect of CDS on fall injury rates in patients aged 80 years or older. There was no statistically significant subgroup difference in fall injury rates when stratified by RoB (p=0.93) or study setting (p=0.41). See S7 and S8 Figs for subgroup analyses on fall injury rate.

## Mortality and hospitalisations

Five CRTs [70–72,84,86] reported on mortality, while five IRPGTs [74–76,82,83] and three CRTs [71,72,86] reported on hospital admissions. The sample sizes were small, the outcomes were rare, and no consistent patterns were observed in the direction of effect for either mortality or hospital admissions. For further details on individual study results, see S8 Table.

**Certainty of the evidence**

The certainty of the evidence was assessed for nine outcomes, with the results summarised in the GRADE evidence profile in S9 Table and Table 4.

## Discussion

This systematic review summarised 25 studies (28 publications) investigating healthcare practitioners' use of CDS in fall prevention and its effects on healthcare practitioner performance and patient outcomes. The interventions were delivered by nurses, physicians, physiotherapists, pharmacists, occupational therapists, and paramedics. Regarding healthcare practitioner performance, CDS may improve fall risk assessments and the provision of recommended interventions and likely improves medication review and prescribing. Regarding patient outcomes, CDS likely decreases the rate of falls in hospitals and residential care settings. CDS also likely reduces falls in patients aged 80 years or older. Furthermore, CDS may reduce fall rates in community-dwelling older adults. It also appears to reduce fall rates in patients aged 65–80 years; however, the effects in these subgroups may be small. CDS interventions likely reduce fall risk slightly and may reduce the rate of fall injuries in patients aged 65–80 years. Finally, while CDS may reduce fall injuries in adults aged 65–80 years, the effect on fall injuries in adults aged 80 years or older remains very uncertain.

### Interventions

A common feature of all experimental interventions included in this review is that they delivered decision support to healthcare practitioners aiming to prevent falls in older adults. The control interventions were generally described only as consisting of usual care, with limited detail about the specific interventions, which varied depending on the setting and type of healthcare practitioner. Most experimental interventions utilised CDS to assist healthcare practitioners in conducting risk assessments and implementing preventive measures across multiple domains, such as gait and balance problems, environmental factors, and medications, rather than focusing on a single domain. Multifactorial fall prevention interventions have been shown effective in reducing falls [58], and the use of CDS to deliver such complex interventions may offer healthcare practitioners structure and guidance [89]. However, it is worth noting that, while most interventions provided CDS electronically, only 36% of the CDS tools automatically delivered recommendations to the healthcare practitioners. A systematic review previously found that automatically providing CDS recommendations, as opposed to requiring practitioners to access them on demand, may lead to large improvements in adherence to clinical guidelines [21].

### Healthcare practitioner performance

Our findings align with those of previous systematic reviews, which have reported improvements in adherence to recommended practice [30,32] and medication outcomes [31]. Mebrahtu et al. [29] also found favourable effects on healthcare practitioner performance, including improvements in nurses' adherence to hand disinfection guidance, insulin dosing, timely blood sampling, and documentation of care. Similarly, Kwan et al. [32] reported that CDS increased the proportion of patients receiving recommended care. Additionally, Yourman et al. [31] found that CDS improved medication outcomes in most cases. Although improvements were mostly moderate, these results demonstrate the diversity of clinical areas in which CDS may be beneficial.

### Patient outcomes

Our meta-analyses suggested a possible reduction in fall risk with CDS, corresponding to an estimated range of 32 fewer to three more fallers per 1,000 older adults. They also indicated that CDS may reduce fall rates, with a reduction ranging from 20 to 188 fewer falls per 1,000 person-years. The studies included in the meta-analysis on fall risk represent a range of age groups and care settings, suggesting that the findings are broadly applicable to older adults at risk of falls. Several

factors may explain the observed differences in the effects of CDS on fall risk versus fall rate. Dautzenberg et al. [11] proposed that the fall outcome may be more accurately measured using fall rate rather than fall risk. The 'fall risk' outcome counts the number of individuals who experience at least one fall, regardless of whether they fall multiple times, with each person contributing only one event. In contrast, the 'rate of falls' outcome captures each individual fall as a separate event. For example, a person who falls five times during the follow-up period would contribute five events to the 'rate of falls' outcome but only one event to the 'fall risk' outcome. If an intervention successfully prevents two of these five falls, it would lead to a reduced fall rate but not a reduced fall risk, as the individual would still be classified as having fallen. Consequently, the 'fall risk' outcome does not capture changes in the frequency of falls among older adults. It is possible that the interventions studied were more effective at reducing the frequency of falls among individuals with the highest fall risk, i.e., recurrent fallers, than among those who experienced only a single fall. A reduction in the frequency of falls among recurrent fallers would result in a greater reduction in fall rate compared to fall risk. In agreement with our findings, earlier systematic reviews [11,90] reported that multifactorial interventions were associated with a reduction in fall rate but had a smaller impact on fall risk.

The studies included in the meta-analyses on fall rate encompass a range of age groups and care settings, suggesting that the findings are broadly applicable to the entire study population. Several factors may explain the differences in fall rate reduction between settings, i.e., a 26% reduction in hospitals and residential care versus a 3% reduction in community-dwelling older adults. Older adults in hospitals and care homes are at high risk of falls [14], and interventions may be more effective in high-risk populations than in those at lower risk [91,92]. Even if the relative effects were similar, interventions directed at high-risk populations tend to show larger absolute effects than those directed at low- or moderate-risk populations. However, it is important to consider whether the relative effect itself varies across populations with different baseline risk levels, as this could further influence intervention outcomes. Furthermore, CDS tools may be more readily implemented in hospital and residential care settings than in primary care due to the greater complexity of these settings [93]. This may increase healthcare practitioners' fidelity to intervention protocols, thereby increasing intervention effectiveness. Another contributing factor could be the differences in follow-up time across the included studies. The median follow-up time was six months for studies conducted in hospitals and residential care, compared to 12 months for studies involving community-dwelling older adults. The effectiveness of fall prevention interventions may dwindle as time passes [94], potentially because of patients discontinuing their engagement in fall prevention programmes, such as exercise regimens, after the study period [14].To reduce the risk of type I errors and avoid false positive results, we chose not to conduct a subgroup analysis by follow-up time. Moreover, while our results indicate a 26% reduction in fall rate in hospitals and residential care, the substantial heterogeneity ($I^2 = 65\%$) suggests that the observed effects likely vary across studies. This variability may be attributed to differences in study populations, settings, or intervention delivery. Despite this heterogeneity, CDS interventions targeting healthcare practitioners likely reduce fall rates in hospitals and residential care settings, although the effect size may depend on the context.

The certainty of the effects on fall injuries is low for patients aged 65–80 years and very low for patients aged 80 years or older, due to risk of bias and imprecision. The wide CIs for fall injury rates may be explained by the fact that fall injuries are relatively rare compared to falls [62]. Importantly, while the estimated effect of CDS on fall injuries among patients aged 80 years or older was a rate ratio of 1.29, the certainty of the evidence is very low. Notably, no studies outside the subgroup analysis on fall injury rate in this age group reported any negative effects attributed to the CDS interventions. Our findings are consistent with those of previous systematic reviews [29,31]. For instance, Yourman et al. found that while CDS may occasionally result in medication prescribing errors, it generally helps to reduce side-effects and improve patient safety [31].

When implementing innovations into clinical practice, both high-income countries (HIC) and low- and middle-income countries (LMIC) face barriers related to political, social, and cultural factors, as well as resource limitations and healthcare practitioner-related factors [95,96]. While the studies included in this review were conducted in HIC, LMIC face

additional barriers, such as physical challenges like unreliable power supplies and poor internet connectivity, limited access to electronic health records and computers, and human resources constraints, including overburdened staff and insufficient formal training for healthcare workers [96]. These limitations in digital infrastructure could hinder the integration of CDS tools, which often rely on access to electronic health records and other digital systems [89]. To address these challenges, CDS systems need to be adaptable to LMIC contexts by incorporating offline functionality or offering paper-based versions. Additionally, CDS tools should be user-friendly to minimise the workload on already overburdened staff.

A notable aspect of this study is the absence of established MID values for fall risk, fall rate, and fall injury rate. We assessed certainty based on whether the true effect lies on the observed side of the null, using the null effect as the threshold. While some of the findings in this review suggest that the effects of CDS may be small, we did not determine what constitutes a minimal clinically important effect, as this often depends on the context and setting.

In HIC with robust healthcare infrastructure, even small reductions in fall risk or fall injuries may justify the use of CDS interventions. Conversely, in LMIC contexts, where resources such as computers, internet access, and trained personnel are limited, the threshold for what constitutes an important difference may be higher. In these settings, interventions with larger impacts may be prioritised to promote the best use of limited resources. Establishing context-specific thresholds for clinical importance could enable a more nuanced evaluation and better inform decisions on whether the implementation of CDS interventions is justified.

## Implications

Given that many of the included studies targeted participants with an increased fall risk, such as previous fallers or those with specific risk factors, the findings of this review are likely most applicable to populations at higher risk of falls. Our estimates indicate that, on average, CDS interventions directed at healthcare practitioners may prevent 14 older adults per 1,000 from falling, assuming a baseline fall risk of 28.7%. Additionally, CDS interventions may prevent 175 falls per 1,000 person-years in hospitals and residential care, as well as 188 falls per 1,000 person-years in adults aged 80 years or older, assuming a baseline fall rate of 672 falls per 1,000 person-years. However, further research is needed to better understand the significant differences in fall rate reduction between hospitals or residential care settings and community-dwelling older adults. Future studies should aim to identify the factors that contribute to the greater effectiveness of interventions in hospitals and residential care and explore ways to adapt these strategies for community-dwelling older adults.

The findings of this review have important implications for the clinical implementation of CDS in different care settings. In hospital and residential care settings, where older adults often have a higher risk of falls and healthcare practitioners operate in more structured environments, CDS may be more readily implemented. The fall rate reductions observed in these settings suggest that CDS can support healthcare practitioners in conducting comprehensive fall risk assessments and delivering multifactorial interventions. In contrast, the implementation of CDS in community-based settings may face additional challenges, as these contexts often involve delivering services within older adults' homes, which typically have less structure and access to digital infrastructure. These differences highlight the need to tailor CDS tools to the specific demands of each setting. For instance, in community-based settings, CDS tools that fit into existing workflows and provide automated decision-support could encourage greater adoption and adherence. Future research should investigate setting-specific determinants for the successful implementation of CDS, with a view to ensuring that tools are adaptable across diverse clinical environments.

This systematic review provides valuable insight to guide the implementation of CDS tools in fall prevention efforts among older adults. Directing interventions at high-risk populations may offer greater benefits than interventions aimed at low- or moderate-risk individuals. Additionally, CDS tools should be designed to automatically deliver on-screen decision support rather than relying on paper-based or on-demand systems [21]. Notably, only 36% of the CDS interventions included in this review met this criterion, highlighting an opportunity to further improve patient outcomes, adherence to recommended practice, and medication outcomes.

## Strengths and limitations

To our knowledge, this is the first systematic review to provide a comprehensive overview of the effects of CDS used by healthcare practitioners in fall prevention among older adults. A major strength of this review is the use of a thorough and sensitive literature search strategy. Potential non-reporting biases, poor indexing, and other factors make it impossible to know whether all relevant studies were in fact identified [41]. We therefore searched the reference lists of included articles and relevant systematic reviews. Additionally, the funnel plots, along with Egger tests and trim-and-fill analyses, showed no evidence of publication bias for fall risk or fall rate outcomes. This indicates that we have likely included most relevant studies. Furthermore, outcomes related to healthcare practitioner performance were reported so varyingly among the studies that meta-analyses were not possible. We consider it a strength that we conducted vote-counting based on the direction of effect, rather than relying solely on textual descriptions of results [41]. Limitations of vote-counting include that the method provides no information on the magnitude of effects and does not account for differences in the relative sizes of the studies. However, vote-counting enables a statistical analysis of whether there is evidence of an intervention effect. Also, this method may be preferable to a narrative description, in which some results are privileged above others without appropriate justification [41]. Moreover, we performed meta-analyses on fall outcomes and applied the GRADE approach, making explicit judgements about the certainty of the evidence [97].

A limitation of this review is that verification of the collected data by a second reviewer was performed for only five of 25 included studies, raising concerns about potential errors during data collection. However, the selection of each result from each study was discussed and verified with the project statistician to ensure accuracy and consistency. Another limitation is the uncertainty surrounding the specific effects of different CDS tools. For example, while some tools delivered decision support to healthcare practitioners conducting multifactorial risk assessments and interventions [38,84], others focused exclusively on guided medication dosing [39,85]. This variation in tool types and use cases makes it difficult to evaluate the specific effects of different tools or to determine whether certain tools have any measurable impact at all. Furthermore, a limitation of the studies included in this review is that most results were judged as having a high or serious risk of bias. All results from the included non-randomised studies were judged to be at risk of bias due to confounding. The lack of randomisation increases the risk of unequal distribution of confounding factors between intervention groups, as group assignment may be influenced by knowledge of prognostic factors. Moreover, none of the included non-randomised studies employed analysis methods that adequately controlled for all important confounders, such as age, history of falls, sex, or gait and balance impairments. The direction and magnitude of this confounding remain unknown, making it difficult to determine whether the reported effect estimates are greater or lower than the true effect. For the randomised studies, the main concerns for risk of bias were related to deviations from intended interventions, such as crossover effects observed in Peterson et al. [85] and intervention contamination reported in Clemson et al. [69].

## Conclusions

CDS likely improves the performance of healthcare practitioners in fall prevention for certain groups of older adults and in specific care settings. While CDS probably reduces falls and may lower fall injury rates, its effects appear to vary across different subgroups of older adults and care settings. This systematic review provides valuable insights into the role of CDS in supporting healthcare practitioners in fall prevention efforts among older adults. Prioritising resources and targeting interventions toward high-risk patients may yield the greatest impact in reducing falls. To maximise effectiveness, interventions should be sustained over time, and CDS tools should be designed to support improved adherence to recommended practice. Future research on fall injuries should aim to improve precision by increasing the number of participants and extending follow-up periods. In addition, meta-analyses of healthcare practitioner performance outcomes would become possible with standardised and detailed reporting of results. Finally, while the design of CDS tools may positively or negatively affect healthcare practitioners' adherence to recommended practice, further research is needed to identify the specific design elements that contribute to successful outcomes in fall prevention.

## Supporting information

**S1 Table. PRISMA checklist.**
(DOCX)

**S2 Table. Differences between protocol and review.**
(DOCX)

**S3 Table. Electronic searches.**
(DOCX)

**S4 Table. Excluded studies.**
(XLSX)

**S5 Table. Description of interventions and funding sources.**
(DOCX)

**S6 Table. Design factors of CDS.**
(DOCX)

**S7 Table. Risk of bias.**
(PDF)

**S8 Table. Individual study results.**
(DOCX)

**S9 Table. GRADE evidence profile.**
(DOCX)

**S1 Appendix. Data used for analyses of healthcare practitioner performance outcomes.**
(XLSX)

**S2 Appendix. Data used for meta-analyses.**
(XLSX)

**S3 Appendix. Extracted data.**
(CSV)

**S1 Fig. Funnell plot of comparison: CDS interventions vs control on fall risk.**
(PNG)

**S2 Fig. Trim-and-fill analysis of comparison: CDS interventions vs control on fall risk.**
(PNG)

**S3 Fig. Meta-analysis comparing CDS interventions with control on fall rate, subgroup analysis by patients' age.**
(PNG)

**S4 Fig. Funnel plot of comparison: CDS interventions vs control on fall rate.**
(PNG)

**S5 Fig. Trim-and-fill analysis of comparison: CDS interventions vs control on fall rate.**
(PNG)

**S6 Fig. Meta-analysis comparing CDS interventions with control on fall rate, subgroup analysis by risk of bias.**
(PNG)

**S7 Fig. Meta-analysis comparing CDS interventions with control on fall injury rate, subgroup analysis by risk of bias.**
(PNG)

**S8 Fig. Meta-analysis comparing CDS interventions with control on fall injury rate, subgroup analysis by study setting.**
(PNG)

## Acknowledgments

The authors are grateful to help provided by Elisabeth Karlsen and Kari Kalland (OsloMet) for electronic database searches.

## Author contributions

**Conceptualization:** Rune Solli, Nina Rydland Olsen, Linda Aimée Hartford Kvæl, Stijn Van de Velde, Are Hugo Pripp, Therese Brovold.

**Data curation:** Rune Solli, Are Hugo Pripp, Therese Brovold.

**Formal analysis:** Rune Solli, Are Hugo Pripp, Signe Agnes Flottorp, Therese Brovold.

**Methodology:** Rune Solli, Nina Rydland Olsen, Linda Aimée Hartford Kvæl, Stijn Van de Velde, Are Hugo Pripp, Signe Agnes Flottorp, Therese Brovold.

**Project administration:** Rune Solli.

**Supervision:** Nina Rydland Olsen, Linda Aimée Hartford Kvæl, Therese Brovold.

**Writing – original draft:** Rune Solli.

**Writing – review & editing:** Rune Solli, Nina Rydland Olsen, Linda Aimée Hartford Kvæl, Stijn Van de Velde, Are Hugo Pripp, Signe Agnes Flottorp, Therese Brovold.

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
