## [Decision Letter · Decision Letter 0]

7 May 2025

Dear Dr. Solli,

We look forward to receiving your revised manuscript.

Kind regards,

Nishant Premnath Jaiswal, MBBS, PhD

Academic Editor

PLOS ONE

2. We note that you have contradictory statements about the literature search in your manuscript. Please can you remove the list of languages the studies reported are in as this suggests that a language restriction was imposed.

3. As required by our policy on Data Availability, please ensure your manuscript or supplementary information includes the following:

Additional Editor Comments (if provided):

Reviewers' comments:

Reviewer's Responses to Questions

**Comments to the Author**

1. Is the manuscript technically sound, and do the data support the conclusions?

Reviewer #1: Yes

Reviewer #2: Yes

2. Has the statistical analysis been performed appropriately and rigorously?

Reviewer #1: Yes

Reviewer #2: Yes

3. Have the authors made all data underlying the findings in their manuscript fully available?

Reviewer #1: Yes

Reviewer #2: Yes

4. Is the manuscript presented in an intelligible fashion and written in standard English?

Reviewer #1: Yes

Reviewer #2: Yes

Reviewer #1: Abstract:

Please clarify the statement that the effect on medications was uncertain. Methods did not describe a search for effects on any medication. Page 10 of the manuscript describes monitoring use of "fall-risk increasing drugs”. It may help to mention this in the abstract so readers know what type of medications were measured/reported.

The statement, “CDS interventions may reduce fall injury rate in older adults aged between 65 and 80 years (RaR 0.80; 95% CI 0.59, 1.09)” does not seem accurate because the confidence interval crosses the 1.0 value. Please consider restating this

If I understand the data correctly, it may be more accurate to change the last sentence of the Abstract to: “but the evidence on fall injury rate in community-dwelling patients aged 80 years or older was very uncertain.” In other words, CDS use worked well in hospitals and residential care, but the effect was not statistically significant in community dwelling patients. This is such an important observation that it needs to be clear here and in the Conclusion of the Abstract on Page 3.

Manuscript (MS)

Please check grammar throughout the MS. For example, Page 4, Introduction line 2 “leading” should be “leads”. This is an important, well-done study, potentially affecting important clinical and health economic outcomes. Grammatical errors distract from its credibility and potential clinical use.

I do not have time to address every grammatical error. This needs review by a good editor.

Reviewer #2: This systematic review and meta-analysis is an important contribution to the literature on fall prevention in older adults using CDS interventions.

The methodology is rigorous, and the conclusions are generally well-supported by the evidence.

Strengths include the comprehensive search, transparent methods, proper use of meta-analytic techniques, subgroup analyses, and the application of GRADE.

Minor points for improvement:

Discussion clarity: Some explanations regarding the difference between fall rate and fall risk outcomes could be streamlined to enhance reader understanding.

Graphical presentation: Figures could be slightly improved to enhance readability (e.g., legends and risk of bias charts could be made larger for clarity).

No ethical concerns were identified.

No concerns regarding plagiarism, redundant publication, or data fabrication were noted.

Recommendation:

Minor revision (language polishing in Discussion and minor graphical improvements).

**Do you want your identity to be public for this peer review?** For information about this choice, including consent withdrawal, please see our Privacy Policy

Reviewer #1: **Yes: ** Laura Bolton, PhD

Reviewer #2: No

---

## [Author Response · Author response to Decision Letter 1]

9 Jun 2025

Response to Reviewers

Dear Dr. Nishant Premnath Jaiswal

Thank you for giving us the opportunity to submit a revised draft of our manuscript titled "Effectiveness of clinical decision support in fall prevention among older adults: a systematic review and meta-analysis" to PLOS ONE. We appreciate the time and effort you and the reviewers have invested in providing valuable feedback on our manuscript. We have carefully considered the comments and have revised the manuscript to address the suggestions provided.

All authors have reviewed and approved the submission of the revised manuscript. The manuscript has not been published and is not being considered for publication elsewhere, in whole or in part, in any language. We hope you will now be able to accept the article for publication in your journal.

Yours sincerely

Rune Solli on behalf of the authors

Response to the Academic Editor

Comment 1:

Response 1:

Thank you for your feedback. We have made changes to the manuscript to meet PLOS ONE’s style requirements, including the following:

• We corrected an error on the title page; please refer to page 1, line 10 for details. We have also updated affiliation 5 and removed affiliation 6 for a co-author. Affiliation 6 was removed because this co-author is no longer associated with that institution. Please see page 1, lines 5, 15, and 16.

• Headings:

o We used level 1 heading for all major sections of the manuscript, level 2 heading for sub-sections of major sections, and level 3 heading for sub-sections within level 2 headings.

o We used bold type, sentence case, and the correct font size for the different heading levels.

• Figures and tables:

o We made corrections to the table and figure titles and legends.

o We reformatted Table 4 to a figure, more specifically to Figure 2, because it contains graphics.

o We renamed Fig 2 as Fig 3, Fig 3 as Fig 4, and Fig 4 as Fig 5.

o We reformatted the manuscript’s figure files as .tif files.

o We renamed Figure file names to Fig1, Fig2, Fig3, etc.

o We changed the resolution of the manuscript Figs to 300 dpi.

o We moved the tables and figures to appear directly after the paragraph where they are first cited.

• Supporting information:

o Please see our response to Comment 3 for details on the changes made to the supporting information.

• References:

o We changed reference style to comply with PLOS ONE’s guidelines. For more information on the changes made to the reference list, please refer to our response to Comment 4.

Comment 2:

We note that you have contradictory statements about the literature search in your manuscript. Please can you remove the list of languages the studies reported are in as this suggests that a language restriction was imposed.

Response 2:

Thank you for pointing this out. We have removed the list of languages the studies are reported in. Please see page 7, lines 154-155.

Comment 3:

As required by our policy on Data Availability, please ensure your manuscript or supplementary information includes the following:

Response 3:

Thank you for the reminder. We have updated the manuscript to include the required information regarding PLOS ONE's policy on Data Availability. We have now included a numbered table listing all studies identified in the literature search, including those excluded from the analysis and the reason for their exclusion. Please see S4 table. One of the included studies, Byrne 2005, is a PhD thesis, and we have now provided a URL link to this source. We have now also included an appendix (S3 Appendix), showing data extracted from the primary research sources, including the name of the data extractors and the date of data extraction, and confirmation that the study was eligible to be included in the review. All the data needed to replicate our analyses were obtained from publicly available primary sources, including journal articles and clinical trials registries. These data are available in S1 and S2 appendices. We have now included the completed risk of bias assessments for each study, along with the answers to the signalling questions, available in S7 Table. We have included a table showing the certainty assessments for each of each outcome, available in S9 table. We explained how missing data were handled, including seeking information from original study authors and imputing missing data. For further details, please see page 12, lines 264-267.

Comment 4:

Response 4:

Thank you for pointing this out. We have updated the references in line with PLOS ONE's guidelines. We have reviewed the references with regards to volume number, page numbers, and article identifiers. We removed Montero-Odasso et al. 2021, titled ‘New horizons in falls prevention and management for older adults: a global initiative,’ from the list of included studies, as it was mistakenly included and did not meet our inclusion criteria. When reviewing the references we identified two duplicate references and these were removed (Tamblyn et al. 2012 and Elley et al. 2008). Additionally, we identified that we had referenced multiple versions of the Cochrane handbook. Therefore, we removed references to Cochrane Handbook version 6.3 from 2022 and earlier versions, and we now cite the correct version that we used, version 6.4 from 2023. We did not cite any retracted articles.

Response to Reviewer 1

Comment 1:

Abstract:

Please clarify the statement that the effect on medications was uncertain. Methods did not describe a search for effects on any medication. Page 10 of the manuscript describes monitoring use of "fall-risk increasing drugs”. It may help to mention this in the abstract so readers know what type of medications were measured/reported.

The statement, “CDS interventions may reduce fall injury rate in older adults aged between 65 and 80 years (RaR 0.80; 95% CI 0.59, 1.09)” does not seem accurate because the confidence interval crosses the 1.0 value. Please consider restating this

If I understand the data correctly, it may be more accurate to change the last sentence of the Abstract to: “but the evidence on fall injury rate in community-dwelling patients aged 80 years or older was very uncertain.” In other words, CDS use worked well in hospitals and residential care, but the effect was not statistically significant in community dwelling patients. This is such an important observation that it needs to be clear here and in the Conclusion of the Abstract on Page 3.

Response 1:

Thank you for pointing this out. We have now added details of the primary outcomes of the systematic review in the methods section of the abstract, including medication outcomes, which consisted of medication review and prescribing. Please see page 2, lines 42-44. In the results section of the abstract, we replaced the statement, “however the effect on medication outcomes was very uncertain,” with “…and medication review and prescribing (all nine comparisons favouring CDS; 95% CI 70%, 100%; low certainty).” Please see page 2, lines 51-53.

Regarding the statement, “CDS interventions may reduce fall injury rate in older adults aged between 65 and 80 years (RaR 0.80; 95% CI 0.59, 1.09),” GRADE guidance suggests that findings should be communicated based on the point estimate and the certainty of evidence [1]. Our best estimate indicates a 20% reduction in fall injury rate, which, together with the low certainty of evidence for this outcome, supports our decision to keep the original phrasing. We hope this explains our choice. If the reviewer still feels this interpretation is incorrect, we are open to reconsider and making necessary changes. We have now clarified in the methods section that results are described in line with GRADE guidance, including a brief explanation of the approach, along with the Santesso reference [1]. Please see page 13, lines 279-281. We have now also included the certainty of evidence rating when reporting the results in the results section of the abstract. Please see pages 2-3, lines 49-61.

Thank you for highlighting the results of CDS interventions among community-dwelling older adults and the uncertainty regarding to fall injuries. We appreciate your insight and have revised the conclusion section of the abstract to emphasize the finding that CDS interventions may reduce falls in hospitals and residential care settings, but likely not in community-dwelling older adults. We have also emphasized the uncertainty of CDS interventions’ effects on fall injury rates in adults aged 80 years or older. Please see page 3, lines 63-70.

Comment 2:

Manuscript (MS)

Please check grammar throughout the MS. For example, Page 4, Introduction line 2 “leading” should be “leads”. This is an important, well-done study, potentially affecting important clinical and health economic outcomes. Grammatical errors distract from its credibility and potential clinical use.

I do not have time to address every grammatical error. This needs review by a good editor.

Response 2:

Thank you for your feedback. We have corrected the sentence in the second line of the introduction; please see page 4, lines 92-93. Additionally, we have reviewed the manuscript and addressed the grammatical errors we identified.

Response to Reviewer 2

Comment 1:

This systematic review and meta-analysis is an important contribution to the literature on fall prevention in older adults using CDS interventions.

The methodology is rigorous, and the conclusions are generally well-supported by the evidence.

Strengths include the comprehensive search, transparent methods, proper use of meta-analytic techniques, subgroup analyses, and the application of GRADE.

Minor points for improvement:

Discussion clarity: Some explanations regarding the difference between fall rate and fall risk outcomes could be streamlined to enhance reader understanding.

Graphical presentation: Figures could be slightly improved to enhance readability (e.g., legends and risk of bias charts could be made larger for clarity).

No ethical concerns were identified.

No concerns regarding plagiarism, redundant publication, or data fabrication were noted.

Recommendation:

Minor revision (language polishing in Discussion and minor graphical improvements).

Response 1:

Thank you for your positive feedback on our systematic review and meta-analysis. We appreciate your recognition of our rigorous methodology. We have revised the manuscript to better explain the differences between fall risk and fall rate outcomes. Specifically, we have clarified the difference between fall risk and fall rate in the methods section; please see page 9, lines 197-203. We have also clarified how interventions might have a different impact on the rate of falls versus fall risk; please see pages 30-31, lines 612-626. We have enlarged the legends and the risk of bias charts to enhance readability. Please see Figs 2-5.

References

1. Santesso N, Glenton C, Dahm P, Garner P, Akl EA, Alper B, et al. GRADE guidelines 26: informative statements to communicate the findings of systematic reviews of interventions. J Clin Epidemiol. 2020;119: 126–135. doi:10.1016/j.jclinepi.2019.10.014

---

## [Decision Letter · Decision Letter 1]

8 Aug 2025

Dear Dr. Solli,

Thank you for submitting your manuscript to PLOS ONE. After careful consideration, we feel that it has merit but does not fully meet PLOS ONE’s publication criteria as it currently stands. Therefore, we invite you to submit a revised version of the manuscript that addresses the points raised during the review process.

We look forward to receiving your revised manuscript.

Kind regards,

Nishant Premnath Jaiswal, MBBS, PhD

Academic Editor

PLOS ONE

Journal Requirements:

Reviewers' comments:

Reviewer's Responses to Questions

**Comments to the Author**

Reviewer #2: (No Response)

Reviewer #3: All comments have been addressed

Reviewer #4: (No Response)

2. Is the manuscript technically sound, and do the data support the conclusions?

Reviewer #2: Yes

Reviewer #3: Partly

Reviewer #4: Partly

3. Has the statistical analysis been performed appropriately and rigorously?

Reviewer #2: Yes

Reviewer #3: Yes

Reviewer #4: Yes

4. Have the authors made all data underlying the findings in their manuscript fully available?

Reviewer #2: Yes

Reviewer #3: Yes

Reviewer #4: Yes

5. Is the manuscript presented in an intelligible fashion and written in standard English?

Reviewer #2: No

Reviewer #3: Yes

Reviewer #4: Yes

Reviewer #2: This manuscript addresses an important and timely topic: the effectiveness of Clinical Decision Support (CDS) systems in fall prevention among older adults. The systematic review and meta-analysis are well designed, and the authors follow rigorous methods aligned with PRISMA and GRADE guidelines. However, several aspects require major attention before the manuscript can be considered for publication... Language and Grammar (Essential Revision):

The manuscript still contains numerous grammatical and syntactical issues that reduce clarity and weaken its impact. A professional English language editing is essential, especially in the Introduction and Discussion sections. For example:

Line 93: “Falls can cause significant health loss that leads to more complex care requirements” → awkward phrasing.

Line 92: “Falls are a major cause of morbidity and mortality…” → avoid redundancy by rephrasing the following lines more concisely.

GRADE Interpretation (Overstatement):

The authors frequently describe outcomes with low or very low certainty as "may improve" or "likely reduce." While this aligns with GRADE semantics, the consistent use of optimistic phrasing throughout the abstract and conclusion may mislead readers about the strength of evidence. I strongly recommend a more cautious tone, especially for:

Medication outcomes: classified as “low certainty” but still interpreted positively.

Fall injury in those 65–80 years: RaR 0.80 with CI crossing 1.00 should be clearly described as statistically non-significant.

Methodological Transparency:

The authors claim no language restriction, but earlier versions included a list of included languages. Ensure this inconsistency is corrected in both the Methods and Appendices.

Risk of bias assessments are now available in S7 Table, but summary RoB graphs in the main text are difficult to interpret (small font, unclear legends). Please improve readability and ensure consistency between main text and supplements.

Data Completeness (Minor):

The explanation for handling missing data is too brief. Please detail the imputation procedures used and specify how often author contact led to data retrieval.

Figures and Tables:

Figure readability is still suboptimal despite the revisions. Font sizes and labeling in forest plots and risk-of-bias visuals should be increased for clarity, especially for print versions.

Minor Points:

Use consistent terminology: sometimes "CDS systems" is used redundantly (the "S" already stands for "system").

Add a visual summary table of GRADE certainty per outcome (possibly alongside S9 Table) in the main manuscript for quick reference.

Discuss potential implementation barriers to CDS in LMIC settings – currently the review is biased toward high-income contexts.

Overall Recommendation:

Minor Revision – Language and Presentation Improvements Required.

Reviewer #3: Thank you for acknowledging the reviewers comment and your revisions. The paper looks better, but I still have a number of issues/questions:

Pg. 5 ln 118 - Suggest changing to "Generally, the use of CDS by healthcare ...."

Pg 5 ln 121 - suggest changing to "patients [32], medication outcomes in older adults [33,34], and adults in general.

Pg 6 ln 134 - suggest changing to "This systematic review aimed to evaluate the effects of CDS for...."

Pg 8 ln 172 - since your inclusion criteria included restrictions on the type of studies for inclusion in the review, please clarify why you did not include restrictions on the study design within your search strategy?

Pg 10 ln 208 - Why did you include studies which did not specify one of your required outcomes?

Pg 14 ln 301 - what do you mean by the phrase "to a median (min-max) of 1 433 (312-46245) patient participants?

Pg 24 ln 460 - your interpretation of the impact of CDS on falls in hospital/residential care, although was significant, needs to include the considerable heterogenity present.

Pg 24 ln 466-478 - your interpretation of the sub group analysis I feel is inaccurate and likely do not reduce the rate of falls.

Pg 28 ln 572-574 - your results do not reflect that CDS does not reduce falls in pts >80

Pg 33 - Strengths and limitations section - please expand more on limitations of the review

Pg 34.

Reviewer #4: Reviewer comments

Thank you for the opportunity to review this revised manuscript. I did not review the original version but have examined the responses to the previous reviewer comments and have reviewed the revised version anew from my perspective.

I did not verify that all requests by the Academic Editor were addressed, but notice that the author did not add any information into the manuscript, as stated in their response, about any methods for author contact for missing outcome data. If they did perform author contact, the means (eg email first author) and frequency should be stated.

Responses to reviewers 1 and 2. The comments to the previous reviewer comments are mostly addressed, though as indicated below more work should be done to define the author’s GRADE approach.

The review is well done and will be improved in my opinion with some minor changes and consideration of one major comment on GRADE. My comments:

Minor comments:

Abstract: i) please note the search date(s), ii) if space permits it would be excellent to provide a brief overview of the types of CDS interventions used most, iii) please see below for some suggestions and questions about the GRADE approach which may influence the reporting of results in the abstract

Introduction: stating that “interventions to prevent falls” are cost-effective and effective in falls and fall injuries could be more specific especially since not all interventions have been shown successful and the authors are arguing for a review of CDS interventions. Possibly “a variety of interventions aimed at older adults” or such would help.

Materials and methods:

1. Was the protocol only ever “drafted”? even if there were changes post hoc the protocol should have been in some form of final version before starting the review.

2. Spell out PRISMA at first use

3. Selection process: please add details about who conducted full text review, and confirm whether consensus was required at title/abstract stage or just full text.

4. Data collection process: using verification for only 5 of 24 papers should be noted in the discussion as a limitation of the review (possibility for errors). Duplicate extraction or verification of all data is a standard for systematic reviews. This is particularly concerning if only one person chose which outcome data to use when studies reported on multiple outcomes.

5. Study risk of bias: please confirm if “the main outcome results” means the result data used for each outcome of interest to the review, or otherwise.

6. Synthesis methods: please add a section on unit of analysis issues, i.e. what was done to account for effects of clustering in cluster RCTs? Was the authors’ result adjusted for clustering used for analysis and, if not reported/performed, what was done?

Results/conclusions:

1. it would be of interest to comment on how many studies had eligibility criteria related to increased risk for falls (e.g. previous fallers, 1+ risk factor etc), and to state whether the findings overall are most applicable to populations at increased risk (this may vary by outcome to some degree),

2. it is hard to know whether the findings for the fall risk outcome may be most applicable to certain settings or ages; can the authors comment on whether the 10 studies appeared to capture both settings and ages fairly well? For example, if the large majority of the participants in the analysis came from studies undertaken in residential/acute care there could be some concerns about directness to the entire population of interest (eg rating down for indirectness may apply) or some way to clarify this observation for the reader,

3. likewise with setting and age groups for fall rates is there any indication that the studies in residential/acute care also enrolled older people to help know whether findings could be most applicable to both setting and age combined?,

4. conversion of ORs to absolute risk difference (for falls risk) should first convert the OR to a RR (see appendix 3 in https://www.bmj.com/content/389/bmj-2024-081904),

5. the authors should speak to their reason for use of a separate study (ref 88) for control event rates (rather than the studies themselves) and describe this study in some detail,

6. in figure 2 there are some “FU” and “follow-up” so this could be made consistent; further the authors should be able to put something into the figure for the medication outcome findings for the Blalock study, even if just a narrative statement by the authors about the direction of effect,

7. for mortality and hospitalizations the reversion to relying on statistical significance goes against what was done for other outcomes; perhaps the authors comment about any directions of effect, speak to the small sample sizes for these rare outcomes and point again to where folks can find the results if they want.

Line 561-2: would suggest deleting this sentence since the authors did not assess the magnitude of effects in their review

Major comment:

GRADE does not asses the certainty of the point estimate per se but it’s relation to the “target” of certainty which should be defined in the methods section (see GRADE 34 guidance and J Clin Epidemiol. 2017 Jul:87:4-13. doi: 10.1016/j.jclinepi.2017.05.006 and BMJ. 2025 Apr 29:389:e081904. doi: 10.1136/bmj-2024-081904). For the provider outcomes, the target would need to be a direction of effect only since the analysis did not assess magnitude. For the patient outcomes, this could have been the null (direction of effect) or some form of a minimally important difference (even if approximate), especially to help determine whether findings are “little to no difference” or “an affect”. It appears the authors just used direction of effect and if so this should be mentioned and the findings should be stated in this light. It is confusing what the authors have done for this since there is no statement in the methods and for falls risk (OR 0.93, 21 fewer fallers per 1000) they state little to no effect in the results (implying 21 fewer is below some threshold of an MID) but a slight reduction in the abstract and discussion.

The choice of the target can change the GRADE ratings for all domains but in particular inconsistency and imprecision. For the provider outcomes, rating a conclusion of direction of effect where all studies showed the same direction would suggest against rating down for inconsistency; the size of effect doesn’t matter for this target. With some of the effects not being statistically significant/precise for the direction there could be concern over imprecision but it may not be serious. For direction of effect for adherence, I would think low certainty overall seems appropriate with the text indicating rating down twice for ROB. The same applies for medication outcomes, unless the authors want to have a third category of No direction which a couple of the findings indicate (e.g. Blum & Lightbody) and where some inconsistency in direction could be noted. If the authors really think low certainty for the direction is appropriate (vs moderate with just serious ROB) they could consider rating down twice for ROB or possibly once for ROB and once for imprecision (as above) or inconsistency (noting really that a direction was not shown in a couple of cases). For the falls risk outcome, if the authors are rating certainty in direction of effect, then not rating down for imprecision makes sense (assuming they wouldn’t be too strict on the upper limit of the 95% CI crossing the null slightly) and they would conclude there is an effect (possibly mentioning that it may not be important), but if they want to say there is “little to no” difference (implying use of an MID and that the point estimate is below this) then they likely should state what the MID was and make sure the entire 95% CI does not contain this value or else also rate down for imprecision. For the falls rate outcomes, if using a direction of effect the authors should note this in their conclusions whereas if they think the magnitude is at least as large as some small but important effect they may also rate down for either inconsistency (since ~30% of the weight in the analysis showed effects below what might be considered an MID) or for imprecision if the lower limit of the 95%CI (81 fewer falls) does not surpass their MID.

In summary, having a clear statement about what the authors were rating their certainty in (e.g. null/direction of effect) would be good as well as re-considering their assessments for the provider outcomes in light of this (not rating down for inconsistency). If using the direction of effect their falls risk findings should likely be “an effect”, with a comment that the effects may be small. If MIDs were applied these should be stated with the above considerations added.

**Do you want your identity to be public for this peer review?** For information about this choice, including consent withdrawal, please see our Privacy Policy

Reviewer #2: No

Reviewer #3: No

Reviewer #4: No

---

## [Author Response · Author response to Decision Letter 2]

21 Sep 2025

Response to Reviewers

Dear Dr. Nishant Premnath Jaiswal

Thank you for the opportunity to submit a revised draft of our manuscript titled "Effectiveness of clinical decision support in fall prevention among older adults: a systematic review and meta-analysis" to PLOS ONE. We greatly appreciate the time and effort you and the reviewers have dedicated to providing thoughtful and constructive feedback. We have carefully considered all comments and made revisions to address the suggestions provided.

All authors have reviewed and approved the submission of the revised manuscript. The manuscript has not been published and is not being considered for publication elsewhere, in whole or in part, in any language. We hope you will now be able to accept the article for publication in your journal.

Yours sincerely

Rune Solli on behalf of the authors

Response to Reviewer 2

Comment 1:

This manuscript addresses an important and timely topic: the effectiveness of Clinical Decision Support (CDS) systems in fall prevention among older adults. The systematic review and meta-analysis are well designed, and the authors follow rigorous methods aligned with PRISMA and GRADE guidelines. However, several aspects require major attention before the manuscript can be considered for publication... Language and Grammar (Essential Revision):

The manuscript still contains numerous grammatical and syntactical issues that reduce clarity and weaken its impact. A professional English language editing is essential, especially in the Introduction and Discussion sections. For example:

Line 93: “Falls can cause significant health loss that leads to more complex care requirements” → awkward phrasing.

Line 92: “Falls are a major cause of morbidity and mortality…” → avoid redundancy by rephrasing the following lines more concisely.

Response 1:

Thank you for this comment. In response to your feedback, the manuscript has undergone professional English language editing by a native English speaker. We have carefully revised the text, particularly the Introduction and Discussion sections, to improve clarity and address the issues you identified.

Comment 2:

GRADE Interpretation (Overstatement):

The authors frequently describe outcomes with low or very low certainty as "may improve" or "likely reduce." While this aligns with GRADE semantics, the consistent use of optimistic phrasing throughout the abstract and conclusion may mislead readers about the strength of evidence. I strongly recommend a more cautious tone, especially for:

Medication outcomes: classified as “low certainty” but still interpreted positively.

Fall injury in those 65–80 years: RaR 0.80 with CI crossing 1.00 should be clearly described as statistically non-significant.

Response 2:

Thank you for your comment on the interpretation of outcomes with low or very low certainty. We acknowledge that the repeated use of phrases such as “may improve” or “likely reduce” throughout the abstract and conclusion could potentially convey an overly optimistic impression. To address this, we have carefully reviewed the text and revised it ensure a more cautious and neutral tone, particularly for outcomes with low or very low certainty, to align with your suggestion and avoid the risk of misleading readers. Please see the updated text. We have explicitly described the result on fall injuries in those aged 65 to 80 years as statistically non-significant. Please see page 29, lines 556-557.

Comment 3:

Methodological Transparency:

The authors claim no language restriction, but earlier versions included a list of included languages. Ensure this inconsistency is corrected in both the Methods and Appendices.

Risk of bias assessments are now available in S7 Table, but summary RoB graphs in the main text are difficult to interpret (small font, unclear legends). Please improve readability and ensure consistency between main text and supplements.

Response 3:

Thank you for your comment regarding language restrictions. No language restrictions were imposed on the literature searches. In response to a previous Editor request, we removed the list of languages to avoid the implication that language restrictions had been applied. We have reviewed the methods section and Appendices to ensure consistency throughout. Additionally, we have revised the summary RoB graphs to improve readability, including increasing font sizes and clarifying legends, and to ensure consistency between the main text and supplements.

Comment 4:

Data Completeness (Minor):

The explanation for handling missing data is too brief. Please detail the imputation procedures used and specify how often author contact led to data retrieval.

Response 4

Thank you for your comment. We have expanded the Methods section to clarify that we contacted the corresponding authors of eight reports to obtain missing data, with up to three email attempts per author. Data retrieval was successful in four instances, and no data imputation was performed. Please see page 10, lines 208-213.

Comment 5:

Figures and Tables:

Figure readability is still suboptimal despite the revisions. Font sizes and labeling in forest plots and risk-of-bias visuals should be increased for clarity, especially for print versions.

Response 5:

Thank you for your comment. We have revised the figures to improve their readability in both the main text and supplementary materials, with particular attention to ensuring clarity in print versions. Font sizes and labelling in the forest plots and risk-of-bias visuals have been increased for enhanced clarity.

Comment 6:

Minor Points:

Use consistent terminology: sometimes "CDS systems" is used redundantly (the "S" already stands for "system").

Add a visual summary table of GRADE certainty per outcome (possibly alongside S9 Table) in the main manuscript for quick reference.

Discuss potential implementation barriers to CDS in LMIC settings – currently the review is biased toward high-income contexts.

Response 6:

Thank you for your observation regarding terminology. For clarity, in this manuscript, we have defined “CDS” as “clinical decision support” and have carefully reviewed the text to ensure consistent use and avoid any potential confusion.

We have added a GRADE summary of findings table in the manuscript for quick reference. Please see Table 4.

Thank you for highlighting the important point about implementation in LMIC. We acknowledge that the review predominantly focuses on high-income contexts. To address this, we have added a discussion of potential implementation barriers in LMIC, including limited digital infrastructure, restricted access to electronic health records, overburdened staff, and other challenges. Please see page 35, lines 713-724.

Response to Reviewer 3

Comment 1:

Pg. 5 ln 118 - Suggest changing to "Generally, the use of CDS by healthcare ...."

Pg 5 ln 121 - suggest changing to "patients [32], medication outcomes in older adults [33,34], and adults in general.

Pg 6 ln 134 - suggest changing to "This systematic review aimed to evaluate the effects of CDS for...."

Pg 8 ln 172 - since your inclusion criteria included restrictions on the type of studies for inclusion in the review, please clarify why you did not include restrictions on the study design within your search strategy?

Pg 10 ln 208 - Why did you include studies which did not specify one of your required outcomes?

Pg 14 ln 301 - what do you mean by the phrase "to a median (min-max) of 1 433 (312-46245) patient participants?

Pg 24 ln 460 - your interpretation of the impact of CDS on falls in hospital/residential care, although was significant, needs to include the considerable heterogenity present.

Pg 24 ln 466-478 - your interpretation of the sub group analysis I feel is inaccurate and likely do not reduce the rate of falls.

Pg 28 ln 572-574 - your results do not reflect that CDS does not reduce falls in pts >80

Pg 33 - Strengths and limitations section - please expand more on limitations of the review

Response 1:

Pg. 5 ln 118 - Thank you for the suggestion. We have updated the sentence to: "Generally, the use of CDS by healthcare ..." for improved clarity.

Pg 5 ln 121 - We have revised the text accordingly.

Pg 6 ln 134 - Thank you for your suggestion. We have revised the sentence to: “This systematic review aimed to evaluate the effects of CDS for fall prevention on …”

Pg 8 ln 172 – Thank you for this observation. Our inclusion criteria restricted study types during the screening and selection phases to ensure relevance and rigor. However, we did not restrict study design in the search strategy to maximise sensitivity, as publications may not always disclose the study design.

Pg 10 ln 208 - Thank you for your questions. Studies that did not specify one of our required outcomes were excluded. We have revised Table 1 in the Methods section to clarify this decision and enhance transparency.

Pg 14 ln 301 - This phrase indicates the median number of patient participants included across the studies, with the smallest study having 312 participants and the largest having 46,245. To improve clarity, have revised this sentence to: "… to a median number of 1,433 patient participants, ranging from 312 to 46,245."

Pg 24 ln 460 – Thank you for pointing this out. We have added a sentence in the Results section to acknowledge the substantial heterogeneity and revised the Discussion section to address the considerable heterogeneity and its implications for interpretating the effects of CDS on falls in hospital/residential care settings.

Pg 24 ln 466-478 – Thank you for pointing this out. Considering comments from all reviewers, we have updated the grading of certainty to explicitly reflect the certainty in the direction of effect. Minor adjustments were made in the certainty ratings, and the corresponding statements about the results have been revised for accuracy and clarity. For further details, please see our response to comment 5 from reviewer 4.

Pg 28 ln 572-574 - Thank you for your comment. Our results reflect that CDS interventions reduce the rate of falls among patients aged 80 years or older. In accordance with GRADE methodology, we have used the term “likely” to describe findings supported by moderate certainty evidence. We have retained this wording and clarified this choice in the methods section under “Assessment of certainty of the evidence”. Please see page 14, lines 312-313 and Table 4.

Pg 33 - Strengths and limitations section - Thank you for the suggestion. We have expanded the Strengths and limitations section to include additional limitations, such as the uncertainty surrounding the specific effects of different types of CDS tools and the substantial risk of bias in most of the included studies.

Response to Reviewer 4

Comment 1

Thank you for the opportunity to review this revised manuscript. I did not review the original version but have examined the responses to the previous reviewer comments and have reviewed the revised version anew from my perspective.

I did not verify that all requests by the Academic Editor were addressed, but notice that the author did not add any information into the manuscript, as stated in their response, about any methods for author contact for missing outcome data. If they did perform author contact, the means (eg email first author) and frequency should be stated.

Response 1

Thank you for your comment. We have updated the Methods section to include details about author contact for missing outcome data. Specifically, we clarified that corresponding authors were contacted via email, with up to three follow-up attempts if necessary. Please see page 10, lines 208-211.

Comment 2

Abstract: i) please note the search date(s), ii) if space permits it would be excellent to provide a brief overview of the types of CDS interventions used most, iii) please see below for some suggestions and questions about the GRADE approach which may influence the reporting of results in the abstract

Introduction: stating that “interventions to prevent falls” are cost-effective and effective in falls and fall injuries could be more specific especially since not all interventions have been shown successful and the authors are arguing for a review of CDS interventions. Possibly “a variety of interventions aimed at older adults” or such would help.

Response 2

Thank you for these suggestions. We have added the search dates to the Methods section of the abstract, included a brief overview of the most used types of CDS, and reviewed the reporting of results in the abstract considering your suggestions regarding the GRADE approach. Please see the revised abstract. In the introduction, we have specified the types of fall prevention interventions that are cost-effective and effective in reducing falls and fall injuries. Please see pages 4-5, lines 103-107.

Comment 3:

Materials and methods:

1. Was the protocol only ever “drafted”? even if there were changes post hoc the protocol should have been in some form of final version before starting the review.

2. Spell out PRISMA at first use

3. Selection process: please add details about who conducted full text review, and confirm whether consensus was required at title/abstract stage or just full text.

4. Data collection process: using verification for only 5 of 24 papers should be noted in the discussion as a limitation of the review (possibility for errors). Duplicate extraction or verification of all data is a standard for systematic reviews. This is particularly concerning if only one person chose which outcome data to use when studies reported on multiple outcomes.

5. Study risk of bias: please confirm if “the main outcome results” means the result data used for each outcome of interest to the review, or otherwise.

6. Synthesis methods: please add a section on unit of analysis issues, i.e. what was done to account for effects of clustering in cluster RCTs? Was the authors’ result adjusted for clustering used for analysis and, if not reported/performed, what was done?

Response 3

1. Thank you for your comment. The protocol was drafted, finalised, and registered in PROSPERO before commencing the review. We have clarified this in the Methods section. Please see page 7, lines 152-156.

2. We have spelled out PRISMA at its first mention.

3. Thank you for your comment. We have added details about who conducted full-text review and clarified that consensus was required at both the title/abstract screening stage and the full-text screening stage. Please see page 9, lines 197-201.

4. Thank you for highlighting this issue. We have discussed the limited verification of collected data by a second reviewer as a limitation of the review in the Discussion section to acknowledge the possibility of errors. Additionally, we have clarified that the selection of each result was discussed and verified with the project statistician to ensure correctness and consistency. Please see page 38, lines 784-787.

5. We have specified which outcomes were assessed for risk of bias. Please see page 12, lines 253-254.

6. Thank you for your comment regarding unit of analysis issues. The cluster-randomised trials included in the review accounted for clustering in their original analyses. Additionally, in our syntheses, we ensured that only one outcome from each study was included, thus maintaining independence between studies in the analyses. We have included a sentence in the methods section to clarify this. Please see page 13, lines 291-292.

Comment 4

Results/conclusions:

1. it would be of interest to comment on how many studies had eligibility criteria related to increased risk for falls (e.g. previous fallers, 1+ risk factor etc), and to state whether the findings overall are most applicable to populations at increased risk (this may vary by outcome to some degree),

2. it is hard to know whether the findings for the fall risk outcome may be most applicable to certain settings or ages; can the authors comment on whether the 10 studies appeared to capture both settings and ages fairly well? For example, if the large majority of the participants in the analysis came from studies undertaken in residential/acute care there could be some conce

---

## [Decision Letter · Decision Letter 2]

10 Oct 2025

Dear Dr. Solli,

Thank you for submitting your manuscript to PLOS ONE. After careful consideration, we feel that it has merit but does not fully meet PLOS ONE’s publication criteria as it currently stands. Therefore, we invite you to submit a revised version of the manuscript that addresses the points raised during the review process.

We look forward to receiving your revised manuscript.

Kind regards,

Sascha Köpke

Academic Editor

PLOS ONE

Journal Requirements:

Reviewers' comments:

Reviewer's Responses to Questions

**Comments to the Author**

Reviewer #2: All comments have been addressed

Reviewer #3: All comments have been addressed

Reviewer #4: (No Response)

2. Is the manuscript technically sound, and do the data support the conclusions?

Reviewer #2: Yes

Reviewer #3: Yes

Reviewer #4: Yes

3. Has the statistical analysis been performed appropriately and rigorously?

Reviewer #2: Yes

Reviewer #3: Yes

Reviewer #4: Yes

4. Have the authors made all data underlying the findings in their manuscript fully available?

Reviewer #2: Yes

Reviewer #3: Yes

Reviewer #4: Yes

5. Is the manuscript presented in an intelligible fashion and written in standard English?

Reviewer #2: Yes

Reviewer #3: Yes

Reviewer #4: Yes

Reviewer #2: This manuscript addresses a highly relevant topic, considering the growing prevalence of falls among older adults and the urgent need for effective interventions. The systematic review and meta-analysis presented demonstrate methodological rigor, including a comprehensive search strategy across multiple databases, application of the GRADE approach, and appropriate use of meta-analytic techniques.

Strengths

Comprehensive and transparent search strategy.

Clearly defined inclusion criteria, encompassing different study designs.

Robust synthesis of the data, including both meta-analyses and risk of bias assessment.

Discussion aligned with the presented evidence, highlighting differences between settings (hospital/residential care versus community).

Suggestions for Improvement

Clarity on outcomes: The explanation of the difference between “fall rate” and “fall risk” could be made more concise and accompanied by practical examples to improve readability for a broader audience.

English language polishing: Although the manuscript has been revised, some sentences could be more fluent and precise. A professional language editing service is recommended.

Graphical presentation: Figures and tables have been improved, but further adjustments could increase readability, such as standardizing font size and enlarging legends, especially for risk-of-bias charts.

Practical conclusions: It would strengthen the paper to explicitly emphasize in the discussion the clinical and implementation implications across different settings (primary care versus hospital/residential), to enhance the translation of findings into practice.

Overall, this is a technically sound study with data that support its conclusions. It represents a valuable contribution to the literature on fall prevention in older adults. I recommend acceptance after minor revisions, mainly focusing on clarity of writing, conceptual distinction between outcomes, and minor graphical improvements.

Reviewer #3: Thank you for resubmitting your revised manuscript, and you have taken into consideration the reviewers feedback. I do have some ongoing feedback /questions which is included in the document which is around:

1. The final number of included studies reported in the manuscript is different to that reported in the PRISMA diagram - please clarify the actual number.

2. Your eligibility criteria is confusing - your report only including studies based in healthcare settings, yet you included community dwelling studies?

3. I do not agree with all your interpretations about the results obtained - you indicate that CDS may reduce fall rate across all age groups and settings, yet this is not supported by your results. Please re-evaluate this.

Reviewer #4: Good revision. Only 2 minor comments for revision. 1. for the GRADE rating of injurious falls in the 80+ age category, there should only be one level rated down for imprecision since the effect shows harm (magnitude not relevant) with slight imprecision (0.99 lower CI limit). This won't change the overall GRADE but will add accuracy. 2. in the discussion, the beginning sentence of the patient outcomes section uses one of the 4 estimates for falls rates without a reason. Perhaps just state that the intervention may reduce fall risk (14 fewer) and fall rates (range 20-188 fewer), since you later discuss the differences across subgroups.

**Do you want your identity to be public for this peer review?** For information about this choice, including consent withdrawal, please see our Privacy Policy

Reviewer #2: No

Reviewer #3: No

Reviewer #4: No

---

## [Author Response · Author response to Decision Letter 3]

14 Dec 2025

Response to Reviewers

Dear Dr. Sascha Köpke

Thank you for the opportunity to submit a revised draft of our manuscript titled "Effectiveness of clinical decision support in fall prevention among older adults: a systematic review and meta-analysis" to PLOS ONE. We sincerely appreciate the time and effort you and the reviewers have invested in providing thoughtful and constructive feedback. We have carefully considered all comments and made revisions to address the suggestions provided.

All authors have reviewed and approved the submission of this revised manuscript. We confirm that the manuscript has not been published and is not under consideration for publication elsewhere, in whole or in part, in any language.

We hope that the revisions meet your expectations, and we look forward to the possibility of having our article accepted for publication.

Yours sincerely

Rune Solli on behalf of the authors

Response to Reviewer 2

Comment 1:

This manuscript addresses a highly relevant topic, considering the growing prevalence of falls among older adults and the urgent need for effective interventions. The systematic review and meta-analysis presented demonstrate methodological rigor, including a comprehensive search strategy across multiple databases, application of the GRADE approach, and appropriate use of meta-analytic techniques.

Strengths

Comprehensive and transparent search strategy.

Clearly defined inclusion criteria, encompassing different study designs.

Robust synthesis of the data, including both meta-analyses and risk of bias assessment.

Discussion aligned with the presented evidence, highlighting differences between settings (hospital/residential care versus community).

Suggestions for Improvement

Clarity on outcomes: The explanation of the difference between “fall rate” and “fall risk” could be made more concise and accompanied by practical examples to improve readability for a broader audience.

English language polishing: Although the manuscript has been revised, some sentences could be more fluent and precise. A professional language editing service is recommended.

Graphical presentation: Figures and tables have been improved, but further adjustments could increase readability, such as standardizing font size and enlarging legends, especially for risk-of-bias charts.

Practical conclusions: It would strengthen the paper to explicitly emphasize in the discussion the clinical and implementation implications across different settings (primary care versus hospital/residential), to enhance the translation of findings into practice.

Overall, this is a technically sound study with data that support its conclusions. It represents a valuable contribution to the literature on fall prevention in older adults. I recommend acceptance after minor revisions, mainly focusing on clarity of writing, conceptual distinction between outcomes, and minor graphical improvements.

Response 1:

Thank you for your feedback. We have clarified the explanation of the difference between “fall rate” and “fall risk” in the Methods, Results, and Discussion sections. Please see page 10, lines 214-221; page 24, lines 438-439; and pages 33-34, lines 663-683. Additionally, we have included a practical example to enhance clarity, which can be found on page 33, lines 673-677.

Thank you for your comment regarding the language of the manuscript. As a reminder, the manuscript underwent professional language editing during the previous revision round. However, we have carefully reviewed the text again in this round to further improve fluency and precision. We hope that the revised manuscript meets the required expectations.

Graphical presentation: Thank you for this suggestion. We appreciate your feedback on improving the figures and tables. In the previous two rounds, we increased the font size and enlarged the legends for the risk-of-bias charts. We believe these figures are now clear, suitable for publication, and compliant with the journal’s submission guidelines regarding figures. However, please let us know if there are any specific aspects that still require further adjustments.

Practical conclusions: Thank you for this valuable suggestion. We have revised the discussion section to explicitly emphasize the clinical and implementation implications of our findings across different settings. Please see the updated Discussion section on page 38, lines 782-795.

Response to Reviewer 3

Comment 1:

Thank you for resubmitting your revised manuscript, and you have taken into consideration the reviewers feedback. I do have some ongoing feedback /questions which is included in the document which is around:

1. The final number of included studies reported in the manuscript is different to that reported in the PRISMA diagram - please clarify the actual number.

2. Your eligibility criteria is confusing - your report only including studies based in healthcare settings, yet you included community dwelling studies?

3. I do not agree with all your interpretations about the results obtained - you indicate that CDS may reduce fall rate across all age groups and settings, yet this is not supported by your results. Please re-evaluate this.

Response 1:

1. Thank you for your comment. To clarify, the PRISMA diagram shows that, from the primary literature searches, 45 reports were assessed for eligibility, of which 21 were excluded. Additionally, as described on the right side of the PRISMA diagram, nine reports were assessed for eligibility via other methods, including the Physiotherapy Evidence Database (PEDro), Google Scholar, chain searches, cited reference searches, and conference abstracts, of which five were excluded. As a result, a total of 28 publications from 25 unique studies were included in the review. We confirm that the inclusion of 28 publications from 25 unique studies is accurately reported in both the PRISMA diagram and the manuscript. We kindly ask for further clarification if there is a specific point we have misunderstood. Additionally, we have clarified the meaning of PEDro in the manuscript by writing it out in full upon its first mention.

2. Thank you for highlighting this discrepancy between the eligibility criteria in the main text and Table 1. We have revised the main text to clarify that studies conducted in any healthcare setting, as well as those conducted in the homes of older adults, were eligible for inclusion.

3. Thank you for your feedback. We have re-evaluated our interpretation of the results and revised the manuscript to ensure it does not suggest that CDS reduces fall rates across all age groups and settings. The updated Results section and revised conclusion now reflect this clarification.

Response to Reviewer 4

Comment 1

Good revision. Only 2 minor comments for revision. 1. for the GRADE rating of injurious falls in the 80+ age category, there should only be one level rated down for imprecision since the effect shows harm (magnitude not relevant) with slight imprecision (0.99 lower CI limit). This won't change the overall GRADE but will add accuracy.

Response 1

Thank you for this valuable point. We agree that only one level should be rated down for imprecision, as the confidence interval slightly crosses the null effect. We have made the necessary changes to the manuscript. Please see Table 4 and S9 Table.

Comment 2

2. in the discussion, the beginning sentence of the patient outcomes section uses one of the 4 estimates for falls rates without a reason. Perhaps just state that the intervention may reduce fall risk (14 fewer) and fall rates (range 20-188 fewer), since you later discuss the differences across subgroups.

Response 2

Thank you for pointing this out. We have revised the beginning sentence of the patient outcomes section of the Discussion as suggested, and it now states that our meta-analyses suggested a possible reduction in fall risk (ranging from 32 fewer to three more fallers) and fall rates (ranging from 20 to 188 fewer falls).

---

## [Editor Report · Decision Letter 3]

16 Dec 2025

Effectiveness of clinical decision support in fall prevention among older adults: a systematic review and meta-analysis

PONE-D-25-10237R3

Dear Dr. Solli,

We’re pleased to inform you that your manuscript has been judged scientifically suitable for publication and will be formally accepted for publication once it meets all outstanding technical requirements.

Kind regards,

Sascha Köpke

Academic Editor

PLOS One
---

## [Editor Report · Acceptance letter]

PONE-D-25-10237R3

PLOS One

Dear Dr. Solli,

I'm pleased to inform you that your manuscript has been deemed suitable for publication in PLOS One. Congratulations! Your manuscript is now being handed over to our production team.

Kind regards,

on behalf of

Professor Sascha Köpke

Academic Editor

PLOS One